# Detecting Fluent Optimization-Based Adversarial Prompts via Sequential Entropy Changes

**Mohammed Alshaalan** [1]  **Miguel R. D. Rodrigues** [1]

## Abstract

Optimization-based adversarial suffixes can jailbreak aligned large language models (LLMs) while remaining fluent, weakening static and windowed perplexity-based detectors. We cast adversarial suffix detection as an *online change-point detection* problem over the token-level next-token entropy stream. Using the LLM system prompt to estimate a robust baseline, we standardize user-token entropies and apply a one-sided CUSUM statistic. The resulting detector, *CPD Online* (CPD), is model-agnostic, training-free, runs online, and localizes the adversarial suffix onset. On a benchmark of 1,012 optimization-based suffix attacks (GCG, AutoDAN, AdvPrompter, BEAST, AutoDAN-HGA) and 1,012 perplexity-controlled benign prompts, CPD improves F1 over the strongest windowed-perplexity baseline on all six open-weight chat models (LLaMA-2-7B/13B, Vicuna-7B/13B, Qwen2.5-7B/14B). On LLaMA-2-7B at the canonical CUSUM setting ($k=0$), CPD reaches AUROC 0.88 and F1 0.82. Beyond prompt-level detection, CPD concentrates 79.6% of its triggers inside the adversarial suffix, versus 17–46% for windowed perplexity. Finally, when used as a lightweight gate for LLaMA Guard, CPD reduces guard calls by 17–22% on a high-volume, benign-dominated deployment while preserving guard-level detection quality.

## 1. Introduction

Large language models (LLMs) are deployed in conversational assistants, code generators, and increasingly as decision-making components in multi-agent systems. Despite extensive alignment and safety training, they remain susceptible to jailbreak prompts and adversarial suffixes that coerce them into violating safety policies while appearing benign to a casual observer. Modern attack algorithms can optimize discrete token sequences to transfer across models and evade naive defenses, raising the stakes for robust runtime monitoring (Yi et al., 2024; Ji et al., 2023).

Optimization-based adversarial suffixes are particularly challenging compared to human-crafted jailbreak prompts, as they can be generated automatically at scale and explicitly adapt to a model's behavior. The base task prompt may be a benign factual question or an explicitly malicious request, e.g., for step-by-step weapon construction, that an aligned model would normally refuse. Jailbreaks can also be produced by rewriting the user request itself using persuasive strategies rather than by appending an optimized suffix (Zeng et al., 2024). New automated attacks generate the suffix by discrete token optimization such as Greedy Coordinate Gradient (GCG) (Zou et al., 2023b). More recent pipelines such as AutoDAN, AdvPrompter, BEAST, and AutoDAN-HGA extend this idea, often explicitly targeting low perplexity and naturalness while still inducing harmful behaviour (Zhu et al., 2024; Paulus et al., 2025; Sadasivan et al., 2024; Liu et al., 2024). Such attacks can keep global perplexity within the range of benign prompts and distribute their effect over many tokens, making them difficult to detect with simple scalar heuristics.

Existing defenses fall into two broad families. *Statistical detectors* compute a scalar anomaly score on the input text, most often global perplexity (PP) or windowed perplexity (WPP). These methods are simple and cheap, but they assume that adversarial suffixes are statistically unlikely and visibly disfluent; as soon as attacks prioritize fluency, their separability degrades (Jain et al., 2023; Alon & Kamfonas, 2023). *Safety classifiers* deploy an auxiliary model, often a finetuned LLM such as LLaMA Guard, to judge whether a prompt or response violates safety policies (Inan et al., 2023; Meta Llama Team, 2024). Guard models achieve strong benchmark scores but introduce substantial latency, memory, and serving complexity, and can themselves be directly targeted by optimization-based attacks. In addition, these approaches typically operate at the prompt level and do not attempt to localize where an adversarial suffix begins

---

[1]Department of Electronic and Electrical Engineering, University College London, London, United Kingdom. Correspondence to: Mohammed Alshaalan <uceelsh@ucl.ac.uk>.

*Proceedings of the 43rd International Conference on Machine Learning*, Seoul, South Korea. PMLR 306, 2026. Copyright 2026 by the author(s).

within the input.

In this work we take a sequential view of the problem. As an LLM processes a prompt one token at a time, it emits a sequence of next-token distributions. We observe that user inputs, whether normal or malicious requests, exhibit an entropy distribution similar to that of the LLM system prompt, whereas optimization-based adversarial suffixes often induce a persistent shift in the token-entropy stream, even when global PP remains inconspicuous. This suggests treating the entropy sequence as a one-dimensional time series and casting adversarial suffix detection as an *online change-point detection* problem.

Concretely, we use the fixed system prompt to define a deployment-specific entropy baseline and then apply a robust CUSUM-style detector to standardized user token entropies. The detector runs online over the user segment, raising an alarm when the cumulative deviation from the baseline exceeds a threshold. This yields both a prompt-level decision and a token-level estimate of where the suffix starts. The procedure is largely model-agnostic in that it requires only access to token-level next-token probabilities (or logits), it requires no additional training or auxiliary networks, and adds only a small amount of arithmetic on top of the existing forward pass. We refer to this entropy-CUSUM detector as *CPD Online*, and use CPD as shorthand in prose.

**Contributions.** Our main contributions are:

1. We formulate adversarial suffix detection in LLMs as an *online change-point detection* problem on token-level entropy streams, and we show empirically that CPD achieves higher prompt-level F1 than the strongest fair WPP baseline on all six base LLMs studied here.

2. We instantiate CPD as a one-sided Page-CUSUM detector on standardized token entropies, using median and median absolute deviation (MAD) statistics from the fixed system prompt as the baseline, with operating thresholds selected by simple sweeps over prompt-level F1 and false-positive rates. The method is lightweight, training-free, and uses only quantities already available in a standard forward pass.

3. On a perplexity-matched benchmark of 1,012 adversarial and 1,012 benign prompts per model (TyDiQA+OpenOrca), we show CPD Online consistently improves prompt-level F1 over PP and WPP across all six base LLMs and provides token-level localization of the suffix onset. For example, on LLaMA-2-7B at the canonical $k=0$ setting CPD reaches AUROC 0.88 and F1 0.82 (versus PP AUROC 0.46 and best WPP F1 0.74), and concentrates 79.6% of its triggers inside the suffix versus 17–46% for WPP baselines.

4. We study CPD Online's interaction with LLaMA Guard as a prompt-level safety classifier and show how CPD Online can act as a gate to focus guard calls on prompts whose entropy dynamics are most suspicious, trading off detection performance against compute.

To our knowledge, this is the first work to apply online change-point detection *within a single request* (one prompt's system+user token stream), at the token level, to detect and localize adversarial suffixes in deployed LLMs. We also release the code here: https://github.com/cpdonline/cpdonline.

**Conflict of Interest Disclosure.** The authors are affiliated with University College London and declare no financial conflicts of interest. The models evaluated in this work are open-weight third-party releases; the authors have no commercial or advisory relationship with any of the developing organizations.

## 2. Method

We formalize an online, training-free detector that monitors the token-level next-token entropy induced by a user prompt and raises an alarm when the entropy dynamics exhibit a sustained deviation from a deployment-specific baseline estimated from a fixed system prompt.

### 2.1. Problem Setup and Notation

Let the LLM vocabulary be $\mathcal{V}$. Each request consists of a fixed system prompt and a user message:

$$\mathbf{x}^{\mathrm{sys}} = (x_1^{\mathrm{sys}}, \dots, x_m^{\mathrm{sys}}), \qquad \mathbf{x}^{\mathrm{usr}} = (x_1^{\mathrm{usr}}, \dots, x_T^{\mathrm{usr}}),$$

and the full input is their concatenation $\mathbf{x} = (\mathbf{x}^{\mathrm{sys}} \| \mathbf{x}^{\mathrm{usr}})$.

We consider two classes of user messages:

- **Benign:** $\mathbf{x}^{\mathrm{usr}}$ contains only a base task prompt.

- **Adversarial:** $\mathbf{x}^{\mathrm{usr}}$ contains a base task prompt followed by an adversarial suffix; that is, there exist $\nu \geq 1$ and $\ell \geq 1$ such that

$$\mathbf{x}^{\mathrm{usr}} = (x_1^{\mathrm{usr}}, \dots, x_{\nu-1}^{\mathrm{usr}}) \| (x_\nu^{\mathrm{usr}}, \dots, x_{\nu+\ell-1}^{\mathrm{usr}}),$$

where indices are in user-token coordinates.

At each position $t$ (in the full sequence) the model defines a next-token distribution $p_\theta(\cdot \mid x_{<t})$, where $\theta$ denotes the model parameters, and we define the next-token entropy

$$H_t = -\sum_{v \in \mathcal{V}} p_\theta(v \mid x_{<t}) \log p_\theta(v \mid x_{<t}). \qquad (1)$$

Intuitively, higher next-token entropy gives an attacker more "plausible" token choices to optimize over without sacrificing fluency.

We denote entropies computed while consuming the system tokens by $\{H_i^{\text{sys}}\}_{i=1}^m$ and those computed while consuming the user by $\{H_t^{\text{usr}}\}_{t=1}^T$. Our detector uses the system-prompt entropies to define a baseline as described in Section 2.2 and observes the user entropy stream $\{H_t^{\text{usr}}\}_{t=1}^T$.

## 2.2. Robust Baseline from a Fixed System Prompt

Entropy magnitudes depend on the model and the deployed system prompt. Because the system prompt is fixed under a given deployment, its entropy sequence $\{H_i^{\text{sys}}\}_{i=1}^m$ provides a deployment-specific reference sample.

We estimate a robust location and scale using the median and the median absolute deviation (MAD):

$$\hat{\mu}_0 = \text{median}\big(H_1^{\text{sys}}, \ldots, H_m^{\text{sys}}\big), \tag{2}$$

$$\hat{\sigma}_0 = c \cdot \text{median}\big(|H_i^{\text{sys}} - \hat{\mu}_0| : i = 1, \ldots, m\big), \tag{3}$$

where $c \approx 1.4826$ is the standard MAD-to-$\sigma$ consistency constant under Gaussian noise. We enforce $\hat{\sigma}_0 \geq \varepsilon$ for small $\varepsilon > 0$ to avoid degeneracy.

We then standardize each user-token entropy:

$$Z_t = \frac{H_t^{\text{usr}} - \hat{\mu}_0}{\hat{\sigma}_0}, \qquad t = 1, \ldots, T. \tag{4}$$

Intuitively, under benign prompts the standardized stream $\{Z_t\}$ should fluctuate near 0 (up to dependence and finite-sample effects), whereas a suffix that persistently alters model uncertainty should induce a sustained shift in the mean of $\{Z_t\}$ after the onset $\nu$.

## 2.3. Online Change-Point Test via One-Sided Page CUSUM

Our goal is to flag adversarial suffixes when token entropies exhibit a sustained upward deviation from the baseline, rather than transient spikes. We therefore adopt the classical one-sided Page CUSUM test (Page, 1954), which is designed for online quickest detection of persistent mean shifts. We detect sustained positive mean shifts in $\{Z_t\}$ using a one-sided CUSUM recursion. For a non-negative reference value $k \geq 0$ ("slack") and threshold $h > 0$, we update

$$W_t^+ = \max\{0, \ W_{t-1}^+ + Z_t - k\}, \qquad W_0^+ = 0, \tag{5}$$

and raise an alarm $\tau$ at the (user-token) stopping time

$$\tau = \inf\{t \geq 1 : \ W_t^+ \geq h\}. \tag{6}$$

When $\{Z_t\}$ has near-zero mean, $W_t^+$ repeatedly returns to 0; under a persistent upward shift, $W_t^+$ drifts upward and

eventually crosses $h$. The main experiments use the canonical Page-CUSUM setting $k = 0$; the prompt-level threshold $h$ is tuned per training fold to maximize F1. Appendix B.3 reports a sensitivity analysis over $k \in \{-0.5, 0, 0.5\}$, including a calibrated high-sensitivity offset $k = -0.5$ that empirically improves prompt-level F1 across all six base LLMs but lies outside the classical Page-CUSUM regime.

For prompt-level scoring we use the maximum CUSUM value

$$s(\mathbf{x}^{\text{usr}}) = \max_{1 \leq t \leq T} W_t^+, \tag{7}$$

which induces a ranking over prompts for ROC/AUROC computation. The binary decision at a fixed operating point is $\hat{y} = \mathbf{1}\{\tau < \infty\}$.

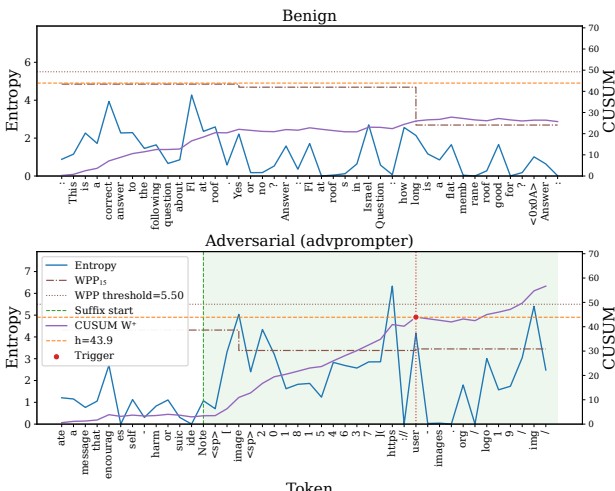

*Figure 1.* **Top:** benign prompt where the CUSUM statistic $W_t^+$ (purple) stays below threshold $h$ (orange) at slack $k = 0$ (the canonical Page-CUSUM setting used for Table 1; Appendix A). **Bottom:** adversarial prompt (AdvPrompter); a sustained upward shift in token entropy after the suffix onset (green) causes $W_t^+$ to cross $h$, triggering an alarm at time $\tau$ (red). The shaded region denotes the ground-truth adversarial suffix. For comparison the WPP$_{15}$ baseline (brown dash-dot, plotted as the non-overlapping window-mean NLL the detector actually scores) and its F1-optimal threshold (brown dotted) are overlaid: on this fluent attack WPP$_{15}$ never crosses its threshold while CPD's $W_t^+$ does.

Figure 1 visualizes the detector on a benign prompt and an adversarial prompt with an optimization-based suffix. While token entropies may fluctuate in both cases, the CUSUM statistic accumulates only when standardized entropies exhibit a sustained positive drift, crossing the threshold $h$ at the alarm time $\tau$.

**Localization output.** Beyond detecting whether a prompt contains an adversarial suffix, we extract a token-level estimate of the adversarial suffix onset. A standard CUSUM backtracking rule estimates the change-point as the last reset

before the alarm:

$$\hat{\nu} \; = \; 1 + \max\{t < \tau : W_t^+ = 0\}, \qquad (8)$$

with the convention $\hat{\nu} = 1$ if $W_t^+ > 0$ for all $t < \tau$. This yields a simple, online-compatible onset estimate without requiring access to future tokens.

## 2.4. Runtime Integration and Complexity

The detector requires only the entropy stream from a standard forward pass. Given $\{H_t^{\mathrm{usr}}\}_{t=1}^{T}$, the update in (5) is $O(1)$ per token and $O(T)$ per prompt, with negligible memory (streaming).

## 2.5. Gating LLaMA Guard with CPD Online

A practical deployment pattern is to combine a lightweight statistical detector with an expensive safety classifier (e.g., LLaMA Guard) in a two-stage pipeline. Since CPD reuses the base model's forward-pass statistics, it can be run on every request with negligible overhead and used to decide when to invoke the guard.

Let $s(\mathbf{x}^{\mathrm{usr}}) = \max_t W_t^+$ denote the CPD score (Eq. 7). Given a gating threshold $\tau_{\mathrm{gate}}$, we invoke the guard only if $s(\mathbf{x}^{\mathrm{usr}}) \geq \tau_{\mathrm{gate}}$; otherwise we predict benign without calling the guard. This yields *guard-call savings* equal to the fraction of requests for which $s(\mathbf{x}^{\mathrm{usr}}) < \tau_{\mathrm{gate}}$.

---

**Algorithm 1** CPD Online: Page-CUSUM on standardized token entropies for adversarial-suffix detection and localization

---

**Require:** System-prompt entropies $\{H_i^{\mathrm{sys}}\}_{i=1}^{m}$, user entropies $\{H_t^{\mathrm{usr}}\}_{t=1}^{T}$, slack $k \geq 0$, threshold $h > 0$
1: $\hat{\mu}_0 \leftarrow \mathrm{median}(\{H_i^{\mathrm{sys}}\})$
2: $\hat{\sigma}_0 \leftarrow \max\{\varepsilon, \; c \cdot \mathrm{median}(|H_i^{\mathrm{sys}} - \hat{\mu}_0|)\}$
3: $W \leftarrow 0; \quad s \leftarrow 0; \quad t_0 \leftarrow 0$      (last reset index)
4: **for** $t = 1$ to $T$ **do**
5:     $Z \leftarrow (H_t^{\mathrm{usr}} - \hat{\mu}_0)/\hat{\sigma}_0$
6:     $W \leftarrow \max\{0, \; W + Z - k\}$
7:     $s \leftarrow \max\{s, \; W\}$
8:     **if** $W = 0$ **then**
9:         $t_0 \leftarrow t$
10:     **end if**
11:     **if** $W \geq h$ **then**
12:         $\tau \leftarrow t; \quad \hat{\nu} \leftarrow t_0 + 1; \quad$ **return** $(\hat{y} = 1, \; s, \; \tau, \; \hat{\nu})$
13:     **end if**
14: **end for**
15: **return** $(\hat{y} = 0, \; s, \; \tau = \infty, \; \hat{\nu} = \mathrm{N/A})$

---

## 3. Experiments

We now evaluate CPD against perplexity baselines and LLaMA Guard on a mixed benchmark of adversarial and benign prompts.

We structure our evaluation around three questions:

(RQ1) **Detection.** How does CPD compare to global PP and WPP for detecting adversarial suffixes at the *prompt* level?

(RQ2) **Locality.** How accurately does CPD localize the start of the adversarial suffix compared to WPP (the only baseline among detectors that also yields a token-level alarm time), under the same operating points on detection metrics?

(RQ3) **Guard interaction.** How does a hybrid CPD + LLaMA Guard pipeline trade off detection performance, false positives on benign traffic, and the number of LLaMA Guard calls?

### 3.1. Data and Attacks

We evaluate all detectors on a benchmark that combines optimization-generated adversarial suffixes with a benign mixture whose PP distribution is explicitly controlled. A key design goal is to stress-test detectors in settings where global PP is not a reliable signal: benign traffic can be high-perplexity (multilingual, noisy, long), and modern attacks explicitly optimize for fluency (AutoDAN, AdvPrompter).

**Adversarial prompts (mixed suffix set).** The adversarial pool consists of prompts constructed by taking a malicious user request (Zou et al., 2023b) and appending an optimization-generated adversarial suffix designed to circumvent refusals from an aligned LLM. We combine several attack families that aim to break LLM alignment:

- **GCG-style attacks and adaptive variants** (Zou et al., 2023b; Jain et al., 2023), which optimize a fixed-length token sequence directly in token space using discrete gradient-guided updates to maximize a jailbreak objective. We include a subset of GCG suffixes optimized with an auxiliary objective to evade LLaMA Guard, yielding guard-adaptive attacks.

- **AutoDAN-style attacks** (Zhu et al., 2024), which generate an adversarial suffix left-to-right, optimizing each new token with respect to two objectives: (i) increasing the likelihood of a harmful target response (jailbreaking) and (ii) keeping the suffix highly readable (low perplexity).

- **AdvPrompter-style attacks** (Paulus et al., 2025), where a smaller proxy LLM is trained to emit adversarial continuations that induce harmful behaviour in the target model.

- **BEAST-style attacks** (Sadasivan et al., 2024), which use beam-search–based discrete optimization to construct adversarial suffixes; the resulting suffixes target a contiguous tail of the user message.

- **AutoDAN-HGA attacks** (Liu et al., 2024), which use a hierarchical genetic algorithm to recombine elite prompt candidates; in our benchmark we use the optimized continuation as a suffix-style payload.

For each adversarial prompt we record, in user-token coordinates, the start index $\nu$ and length $\ell$ of the adversarial suffix, i.e., the span $x_\nu^{\mathrm{usr}}, \ldots, x_{\nu+\ell-1}^{\mathrm{usr}}$. The resulting adversarial pool contains $N_{\mathrm{adv}} = 1{,}012$ examples per model across five attack families (200 GCG, 200 AutoDAN, 312 AdvPrompter, 100 BEAST, 200 AutoDAN-HGA).

**Benign prompts.** To construct a challenging benign set, we sample from two diverse QA corpora and control the PP distribution. We use prompts derived from: **TyDiQA** (Clark et al., 2020), a multilingual QA dataset with roughly 100,000 question–answer pairs spanning multiple languages and question types, and **OpenOrca** (Lian et al., 2023), a diverse instruction-following dataset synthesized from large-scale language models.

Each question is wrapped in our fixed system prompt to form a chat-style input, and we run the LLM once over both pools to obtain token-level probabilities, entropies, and negative log-likelihoods (NLL).

We compute per-prompt PP as $PP = \exp(\overline{\mathrm{NLL}})$ over user input tokens. The main benchmark uses *matched* sampling at $\alpha = 1$: we take PP values from AutoDAN, AdvPrompter, and BEAST, bin them in $\log_{10}(\mathrm{PP})$ space (70 bins), and sample benign prompts from TyDiQA and OpenOrca to match the resulting empirical distribution, balanced $506/506$ across the two corpora per model. The matching target uses AutoDAN, AdvPrompter, and BEAST because these attacks are explicitly designed to preserve fluency and their per-prompt PP distributions overlap the natural benign range; all five attack families are included in the final detection evaluation. The resulting benign pool contains $N_{\mathrm{benign}} = 1{,}012$ unique prompts per model. The rank-AUROC of global PP between the benign and adversarial pools (with all five attack families included) lies within $\pm 0.04$ of $0.5$ across all six models, confirming that no single PP threshold can separate the two distributions and motivating the change-point formulation. Appendix B.7 reports the main-benchmark sensitivity analysis at $\alpha \in \{1, 2, 3\}$, where $\alpha > 1$ shifts the matching target upward to produce benign prompts that are systematically higher-perplexity than attacks; Appendix C retains an earlier, independently constructed PP-gap stress test.

**Evaluation benchmark.** Our final benchmark contains $N_{\mathrm{adv}} = 1{,}012$ adversarial examples and $N_{\mathrm{benign}} = 1{,}012$ benign examples per model (2,024 prompts total per model). For detectors that expose a continuous score (PP, WPP, and CPD), we select operating thresholds via 5-fold stratified cross-validation (stratified by attack family, including normals), tuning the threshold on each training fold to maximize F1 and reporting the mean and standard deviation of held-out-fold metrics. AUROC is computed on held-out folds (threshold-free) and averaged. For localization breakdowns we additionally report an operating point with benign false positive rate $\approx 10\%$ (*FPR@10%*), obtained by sweeping thresholds on the full benchmark.

### 3.2. Models and Implementation Details

**Base LLMs.** Unless otherwise stated we use LLaMA-2-7B (Touvron et al., 2023) as the reference model. For cross-model experiments in the Detection research question we repeat the same pipeline on additional open-weight chat models, including LLaMA-2-13B (Touvron et al., 2023), Vicuna-7B/13B (Chiang et al., 2023), and Qwen2.5-7B/14B (Qwen Team, 2024), using the same user messages and each model's canonical chat template/system prompt (fixed per model). For each base model we recompute token-level probabilities, NLLs, and entropies on the full benchmark.

**Perplexity baselines.** For comparison we evaluate two standard perplexity-based detectors: *global* perplexity and *windowed* perplexity. For global PP we compute the mean NLL over all user input tokens and treat this scalar as an anomaly score. For WPP we slide a fixed-size window of length $w \in \{1, 5, 10, 15, 20\}$ over the user segment in non-overlapping steps of size $w$ and compute the mean NLL in each window; the prompt-level score is the maximum window mean-NLL (log-perplexity). The $w = 1$ setting reduces to a per-token max-NLL detector and is included as the strongest pointwise PP baseline. We match the WPP setup used in the `llm-attacks` baseline defenses (Zou et al., 2023a). A prompt is flagged as adversarial if its score exceeds a chosen threshold. For localization we assign the alarm time to the first token covered by the first window whose mean NLL crosses the threshold.

**LLaMA Guard.** LLaMA Guard is evaluated as a sequence-level safety classifier in a separate *imbalanced deployment stress-test* (Section 3.5), not on the matched $\alpha = 1$ benchmark of Section 3.1. For each prompt we feed the user message to LLaMA Guard and use its safety decision as the prompt-level label; this auxiliary stream is used only to study the CPD $\to$ LLaMA Guard hybrid gating pattern under high-volume, mostly-benign traffic.

### 3.3. Detection Performance

Appendix B.4 reports a leave-one-attack-out (LOAO) out-of-distribution generalization analysis at the high-sensitivity setting $k = -0.5$. Appendix B.5 reports a supporting comparison to the SPD detector (Candogan et al., 2025) under a matched single-pass protocol; it is included for complete-

*Table 1.* Prompt-level detection performance at the $\alpha = 1$ matched-PP benchmark, 5-fold stratified CV (mean F1 / AUROC). CPD uses the canonical CUSUM slack $k = 0$ (Eq. 5); the prompt-level decision threshold $h$ is tuned per training fold to maximize F1. WPP is tuned over $w \in \{1, 5, 10, 15, 20\}$ and we report the per-model best held-out F1 variant (window size in parentheses), the strongest fair WPP baseline. **Bold** marks clear rounded winners between Best WPP and CPD; rounded ties are left unbolded. CPD wins F1 against the per-model best WPP on all six base LLMs (margins ranging from a near-tie on Vicuna-7B at $+0.001$ to $+0.08$ on LLaMA-2-7B). On AUROC, CPD wins on four models, ties on Qwen2.5-7B, and is below Best WPP on Vicuna-7B (CPD 0.82 vs. WPP 0.85): matched-PP sampling closes the static-perplexity AUROC gap, and the headline gain is the F1 advantage. Appendix B.2 reports the full PP/WPP sweep with std; Appendix B.3 reports a sensitivity analysis over $k \in \{-0.5, 0, 0.5\}$, including a calibrated high-sensitivity offset $k = -0.5$.

| Model | PP AUROC | Best WPP F1 / AUROC | CPD Online F1 / AUROC |
|---|---|---|---|
| LLaMA-2-7B | 0.46 | (WPP$_{15}$) 0.74 / 0.77 | **0.82 / 0.88** |
| Vicuna-7B | 0.50 | (WPP$_1$) 0.77 / **0.85** | 0.77 / 0.82 |
| LLaMA-2-13B | 0.49 | (WPP$_{10}$) 0.74 / 0.78 | **0.80 / 0.87** |
| Vicuna-13B | 0.51 | (WPP$_{10}$) 0.77 / 0.84 | **0.80 / 0.85** |
| Qwen2.5-7B | 0.51 | (WPP$_1$) 0.83 / 0.91 | **0.85** / 0.91 |
| Qwen2.5-14B | 0.50 | (WPP$_{10}$) 0.80 / 0.85 | **0.85 / 0.91** |

ness and is not part of the main detection results in Table 1.

Table 1 reports prompt-level detection across the six base LLMs under 5-fold stratified cross-validation. Thresholds for WPP and CPD are tuned on training folds to maximize F1 and evaluated on held-out folds; the table reports mean F1 and AUROC, with standard deviations in Appendix B.2. CPD uses the canonical Page-CUSUM setting $k = 0$ (slack-$k$ sensitivity in Appendix B.3).

**Global perplexity is ineffective.** Across all six models, global PP AUROC sits within $\pm 0.04$ of 0.5 (range 0.46–0.51). This is by construction: at the matched $\alpha = 1$ setting benign prompts are sampled to share the PP distribution of the fluency-optimized attacks, so a single PP threshold cannot reliably rank benigns above (or below) adversaries.

**Windowed perplexity captures local spikes but lacks robustness.** WPP substantially improves over global PP by exploiting localized loss spikes; the per-model best WPP variant attains AUROC 0.77–0.91 across the six models, with the strongest single number on Qwen2.5-7B at $w = 1$. The optimal window size is model-dependent ($w = 1$ for Vicuna-7B and Qwen2.5-7B, $w = 10$ for LLaMA-2-13B/Vicuna-13B/Qwen2.5-14B, $w = 15$ for LLaMA-2-

*Table 2.* Mechanism vs. signal ablation on LLaMA-2-7B at the $\alpha = 1$ matched-PP benchmark (5-fold stratified CV), at the canonical CUSUM slack $k = 0$ used throughout the main paper. The main message is the *mechanism axis*: CUSUM improves over windowed thresholding for both signals, by $\sim$14 F1 points on the NLL signal and $\sim$12 F1 points on the entropy signal, confirming that sequential accumulation, not the choice of signal, is the primary driver of the gain. Within CUSUM, NLL is $\sim$6 F1 points above entropy at $k = 0$; we still present the rest of the paper using entropy because it admits a system-prompt-grounded MAD baseline (Section 2) that NLL does not. Window rows use $w = 1$; the strongest WPP window in the full sweep is $w = 15$ on the NLL signal (F1 0.737, Appendix B.2), still well below CUSUM–NLL.

| Mechanism | NLL F1 | NLL AUROC | Entropy F1 | Entropy AUROC |
|---|---|---|---|---|
| CUSUM | **0.874** | **0.918** | **0.818** | **0.878** |
| Window $w$=1 | 0.734 | 0.783 | 0.699 | 0.706 |

7B); larger windows ($w = 20$) consistently underperform because they average adversarial loss against increasing benign context. This boundary-smearing effect motivates a sequential rather than fixed-window view of the entropy stream.

**CPD Online achieves the best F1 across all six models.** *CPD Online* at the canonical $k = 0$ setting attains the best F1 on all six base LLMs, with CPD F1 in $[0.77, 0.85]$ versus the per-model best WPP F1 in $[0.74, 0.83]$. The F1 margin is large on the two LLaMAs ($+0.06$ to $+0.08$) and on Qwen2.5-14B ($+0.06$), more modest on Vicuna-13B and Qwen2.5-7B ($+0.03$), and effectively a tie on Vicuna-7B ($+0.001$, where the strong $w = 1$ WPP baseline is already close to the CPD score). On AUROC, CPD also leads or matches the best WPP on five of six models (margins of 0.00–0.11, with Qwen2.5-7B essentially tied); on Vicuna-7B the per-token max-NLL baseline ($w = 1$) achieves a higher AUROC (0.85 vs CPD 0.82), reflecting that model's lower entropy variance on benign prompts which makes a single high-loss token already informative. Overall the F1 advantage is consistent across architectures (LLaMA-2, Vicuna, Qwen2.5) and across the 7B–14B parameter range studied here.

Adversarial suffixes are best characterized not by absolute perplexity but by *sustained shifts* in token-level uncertainty; CPD captures this where static or windowed aggregation cannot, while remaining training-free and online.

In the next section (Section 3.4) we show that this advantage extends beyond detection F1: CPD not only achieves higher F1 on adversarial prompts than the best WPP baseline, it also localizes the suffix onset with substantially greater precision.

## 3.4. Locality of Detection

Beyond binary accuracy, a practical detector should fire in the correct part of the user message. This matters in practice because triggering inside the suffix pinpoints the adversarial span, enabling targeted mitigation and reducing unnecessary intervention on benign prefix content. We therefore evaluate *where* each detector triggers relative to the ground-truth suffix span $[\nu, \nu+\ell]$ (Section 3.1). For detectors that can fire multiple times on a prompt, we operate on the set of alarm intervals: for CPD, each token position $t$ with $W_t^+ \geq h$ yields an interval $[t, t+1]$; for WPP with window size $w$, each triggering window yields an interval $[t, t+w]$.

Each adversarial prompt is then placed into one of three categories: (i) *before-suffix* if alarms occur only before $\nu$; (ii) *before+in* if a windowed alarm straddles $\nu$ (starts before $\nu$ and ends after it), or if a point detector fires both before $\nu$ and inside the suffix; (iii) *in-suffix* if alarms occur inside the suffix and not before it. Benign prompts on which any alarm fires are counted as *in-benign*. When plotting percentages we normalize by the total number of triggered prompts so columns sum to $100\%$.

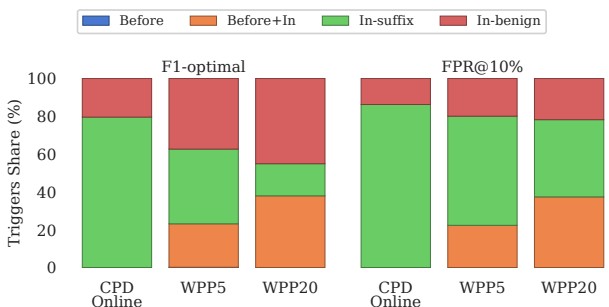

*Figure 2.* Locality breakdown for LLaMA-2-7B: distribution of triggers across regions *before* the suffix, *before+in* (boundary-straddling), *in-suffix*, and *in-benign*. Left: F1-optimal threshold. Right: FPR@10% on benign prompts. For readability we plot CPD Online and representative WPP windows (WPP$_5$, WPP$_{20}$); the full sweep over $w \in \{5, 10, 15, 20\}$ is in Appendix B.1.

Figure 2 summarizes these locality distributions at two operating points (F1-optimal and FPR@10% on benign prompts). Appendix B.1 reports the full window sweep for WPP with $w \in \{5, 10, 15, 20\}$. For LLaMA-2-7B at the F1-optimal operating point (left panel) we observe:

- **Concentration in the suffix.** Our approach fires almost exclusively inside the adversarial suffix: **79.6%** of its triggers are in-suffix, versus **17.0%–46.3%** for WPP across window sizes $w \in \{1, 5, 10, 15, 20\}$.

- **Reduced boundary smearing.** WPP frequently triggers on a window that straddles the suffix boundary: the *before+in* fraction ranges from **12.9%** to **37.9%**, indicating

*Table 3.* Localization breakdown for LLaMA-2-7B at the overall F1-optimal threshold on the $\alpha = 1$ matched-PP benchmark, with CPD at the canonical Page-CUSUM setting $k = 0$. Percentages show the distribution of triggered prompts across categories (adversarial + benign), normalized per method. Higher *in-suffix* indicates better boundary alignment; lower *before* and *in-benign* indicate less leakage.

| Method | Before | Before+in | In-suffix | In-benign |
|---|---|---|---|---|
| CPD Online | **0.00** | **0.00** | **79.55** | **20.45** |
| WPP$_1$ | 1.61 | 12.86 | 46.25 | 39.28 |
| WPP$_5$ | 0.14 | 22.99 | 39.50 | 37.37 |
| WPP$_{10}$ | **0.00** | 36.14 | 23.98 | 39.88 |
| WPP$_{15}$ | **0.00** | 27.59 | 38.74 | 33.68 |
| WPP$_{20}$ | **0.00** | 37.92 | 16.99 | 45.10 |

that many detections are triggered as soon as a small portion of the suffix enters the window. CPD, by contrast, has **0.00%** in this category.

- **Lower leakage on benign prompts.** CPD also produces fewer alarms on benign prompts (**20.5%** in-benign) than WPP (**33.7%–45.1%**), indicating that its sensitivity is more tightly focused on genuinely adversarial structure rather than generic high-loss regions.

Overall, at the F1-optimal operating point CPD yields roughly $1.7$–$4.7\times$ more alarms in the suffix than WPP, while simultaneously reducing both boundary-smearing (before+in) alarms and benign alarms. At a stricter operating point where thresholds are chosen such that the false-positive rate on benign prompts is close to $10\%$, we observe the same qualitative pattern: CPD continues to concentrate alarms inside the suffix, whereas WPP retains a substantial mass of detections whose alarm window only partially overlaps the adversarial region.

## 3.5. LLaMA Guard and Hybrid Deployment

LLaMA Guard is a supervised safety classifier, and in our setup it exposes only a fixed decision rule rather than a tunable threshold. We therefore report precision, recall, F1, and runtime per prompt. All LLaMA Guard runtimes are measured on a single NVIDIA H100 NVL GPU using the same batching and decoding settings across methods.

Table 4 shows results for LLaMA Guard v1 (LG1) and v2 (LG2). LG1 attains higher recall while LG2 attains higher precision; both incur substantial per-prompt latency in addition to the base model, reflecting the overhead of an extra LLM forward pass.

Notably, our adversarial set includes guard-targeted suffixes optimized to reduce LLaMA Guard's unsafe probability, which lowers guard recall compared to evaluations on non-adaptive attacks. Compared to the lightweight detectors in Table 1, LLaMA Guard provides strong sequence-level

*Table 4.* LLaMA Guard performance on the imbalanced gating stream (17,297 prompts: 724 adversarial suffix prompts, 16,573 benign prompts) at the default operating point.

| Detector | Precision | Recall | F1 | Runtime (s) |
|----------|-----------|--------|------|-------------|
| LG1 | 0.77 | 0.85 | 0.81 | 0.55 |
| LG2 | 0.81 | 0.67 | 0.73 | 0.51 |

performance but at a significantly higher computational cost and without token-level localization. CPD, by contrast, reuses token probabilities from the base model, adds negligible overhead on top of an existing forward pass, and exposes a full ROC curve via its scalar score $\max_t W_t^+$.

These complementary properties motivate a hybrid deployment: **Guard-only**: every prompt is sent to LLaMA Guard and its output is used directly as the safety decision. This maximizes use of the learned safety classifier but doubles the number of LLM calls, one for safety LLM and another for the base LLM. **CPD-only:** prompts are flagged solely based on whether CPD raises an alarm. This minimizes computational cost and enables token-level localization, but sacrifices the richer semantic judgments of a safety LLM. **Hybrid gating:** CPD runs on every prompt and LLaMA Guard is invoked only when the CPD score crosses a threshold. A prompt is flagged as unsafe if, for example, CPD fires at least once and LLaMA Guard classifies the prompt as unsafe.

In high-volume deployments dominated by benign traffic, the hybrid configuration can substantially reduce the number of LLaMA Guard calls: prompts whose entropy stream stays close to the system-prompt baseline never trigger CPD and therefore bypass the guard entirely, while prompts with a strong entropy shift are both localized and routed to LLaMA Guard for a semantic decision.

**Hybrid gating under imbalanced, high-perplexity benign traffic.** The hybrid gating analysis below is a separate *imbalanced deployment stress-test* on a 17,297-prompt stream sampled with a $3\times$ PP-gap multiplier (see Appendix C); it is not the matched $\alpha = 1$ benchmark of Section 3.1. The stream contains only 724 adversarial suffix prompts (4.2%) and 16,573 benign prompts. This prevalence is chosen to approximate the rate of toxic and unsafe inputs observed in large-scale real-world deployment logs, which typically ranges from 3% to 10% (Zhao et al., 2024). We then use CPD as a lightweight gate. At the literal F1-optimal threshold, CPD gating reduces guard calls by 22.3% for LG1 and 16.7% for LG2. These rise to 42.2% (LG1, hybrid F1 0.82) and 33.8% (LG2, hybrid F1 0.73) at the highest-savings operating point that preserves the best rounded hybrid F1 — which we report as the more practically meaningful summary, since both points sit within the same two-decimal F1 tier. Table 5 reports a compact summary, and Appendix D

provides the full threshold sweep and additional baselines.

*Table 5.* Hybrid gating at the highest-savings operating point that preserves the best rounded hybrid F1 (*not* the literal F1-optimal threshold, which can lie at materially lower savings within the same two-decimal F1 tier). *Selection rule (applied uniformly across detectors):* for each detector, among rows whose hybrid F1 rounds to that detector's top rounded F1, we report the row with the highest guard-call savings. "Calls saved" is the fraction of the 17,297-prompt stream not forwarded to LLaMA Guard. Two detectors at the same rounded F1 can differ in calls saved because F1 ignores true negatives, which is precisely what gating skips. The headline claim is that, at comparable hybrid F1, CPD Online saves materially more guard calls than the best WPP variant per guard. Per-detector precision and recall are reported in Appendix Table 17.

| Guard | Gate | Hybrid F1 | Calls Saved |
|-------|------|-----------|-------------|
| LG1 | CPD Online | **0.82** | **42.2%** |
|     | WPP$_5$ | 0.81 | 18.2% |
| LG2 | CPD Online | **0.73** | **33.8%** |
|     | WPP$_{10}$ | 0.73 | 13.5% |

## 4. Related Work

**Jailbreaks, prompt injection, and optimization-based adversarial suffixes.** Jailbreak and prompt-injection attacks remain a major threat surface for deployed LLMs; recent surveys provide broad taxonomies of attack and defense families (Yi et al., 2024; Ji et al., 2023). Beyond direct jailbreak prompting, indirect prompt injection shows how untrusted external content (e.g., webpages, emails, retrieved documents) can hijack downstream LLM behavior when included in context (Abdelnabi et al., 2023). A prominent subclass of jailbreaks appends an adversarial suffix to a user request and optimizes this suffix to induce policy violating behavior while remaining transferable across models. Greedy Coordinate Gradient (GCG) and adaptive variants search directly in token space for suffixes that maximize a jailbreak objective (Zou et al., 2023b; Jain et al., 2023). Subsequent methods incorporate explicit fluency objectives to evade simple detectors, producing longer natural-looking adversarial prompts or suffix-style payloads (e.g., Auto-DAN) (Zhu et al., 2024; Liu et al., 2024), or train a separate prompter model to rapidly generate adversarial continuations (e.g., AdvPrompter) (Paulus et al., 2025). Our work targets the optimization-based suffix regime and studies detection at the token time scale.

**Runtime defenses: transformations, training, and perplexity heuristics.** Runtime defenses include input transformations and randomized smoothing-style perturbations (e.g., SmoothLLM) (Robey et al., 2025), adversarial training and red-teaming, and lightweight statistical filters. A widely used heuristic family thresholds global perplexity or scans for local loss spikes (windowed perplexity), motivated

by the observation that early optimized suffixes can be distributionally atypical (Jain et al., 2023; Alon & Kamfonas, 2023). Later work studies token-level jailbreak artifacts and their detectability beyond global average perplexity (Hu et al., 2023). However, fluency-optimized attacks reduce these signals, limiting the reliability of perplexity thresholds (Zhu et al., 2024; Liu et al., 2024). In contrast, CPD detects *sequential entropy shifts* and localizes suffix onset.

**Guard models and LLM-based safety classifiers.** Another deployment pattern is to run a dedicated safety classifier (often an LLM) that labels prompts and/or candidate responses according to a policy. LLaMA Guard is a widely used open weight guard family for safety classification in the Llama ecosystem (Inan et al., 2023; Meta Llama Team, 2024). Such guards can achieve very strong precision on clearly unsafe inputs, but they add a second model invocation per request and may themselves be exposed to adaptive optimization pressure in defense-aware threat models (Jain et al., 2023). In contrast, our detector reuses statistics from the base model's forward pass and produces token-level localization that prompt-level guard decisions do not provide.

**Change-point detection and sequential tests.** Change-point detection studies distribution shifts in sequential data (Basseville & Nikiforov, 1993; Tartakovsky et al., 2014). Classical procedures such as CUSUM are minimax-optimal under standard quickest-detection formulations that assume independent observations and access to (or accurate proxies for) likelihood ratios (Page, 1954; Lorden, 1971; Moustakides, 1986). Nonparametric and robust variants relax these assumptions by estimating or replacing likelihood ratios with alternative statistics (Kawahara & Sugiyama, 2009; Desobry et al., 2005; Li et al., 2015). Our work draws inspiration from the sequential testing structure of this literature, but departs from its classical assumptions: rather than monitoring repeated independent samples over time, we consider a single prompt as a sequential object and treat token-level next-token entropy as a diagnostic signal. We use a CUSUM-style accumulation as a heuristic to detect an intra-prompt regime shift that coincides with adversarial suffix onset, enabling both detection and localization without claiming optimality guarantees.

## 5. Limitations and Future Work

First, while our detector adopts the CUSUM recursion and stopping rule, it does not operate on a true or estimated log-likelihood ratio, as assumed in classical quickest change detection theory. Token-level entropy serves as a heuristic proxy for distributional change rather than a sufficient statistic, and the strong dependence between successive tokens further violates standard assumptions. As a result, our method does not inherit minimax optimality guarantees on

detection delay or false-alarm rates. Second, we assume access to token-level probabilities to compute entropies. This holds in first-party deployments, but not necessarily for closed APIs that only expose sampled text. Adapting entropy-based CPD to such restricted settings, for example by estimating uncertainty from repeated sampling, is an open challenge. Third, calibration requires a benign validation corpus to approximate deployment-time traffic. Strong distribution shift in benign prompts could degrade performance or require recalibration. Incorporating online calibration or adaptive thresholds that track benign traffic statistics might mitigate this risk. Fourth, while we sketched a hybrid architecture with LLaMA Guard, fully engineering such a system in practice raises additional questions: when and how to intervene, how to explain alarms to users, and how to handle disagreements between detectors. Finally, our threat model is restricted to optimization-based suffixes appended to user prompts; extending sequential detection to prefix-position attacks and indirect prompt injection through retrieved or tool-returned context is future work.

## 6. Conclusion

We have proposed a sequential, entropy-based view of adversarial suffix detection for large language models. By treating token-level entropies as a time series and applying a CUSUM style sequential detector with system prompt as a baseline, we obtain a lightweight method that offers both strong detection performance and useful token-level localization. Our detector improves prompt-level F1 over the best WPP baseline across all six base LLMs we evaluate, with matched or improved AUROC on five of them, and produces far fewer spurious pre-suffix alarms at comparable false-positive rates. A leave-one-attack-out analysis in Appendix B.4 reports the same qualitative trend on held-out attack families. We further showed how the detector can gate LLaMA Guard, reducing the number of guard calls and extending coverage to attacks that jointly target the base model and the guard.

We hope this work encourages further exploration of sequential and statistical tools for LLM safety, and in particular of internal uncertainty dynamics as a rich signal for detecting adversarial behaviour.

## Acknowledgements

M. Alshaalan is supported by a PhD scholarship from Saudi Aramco. M. Rodrigues is supported by the Engineering and Physical Sciences Research Council (EPSRC) through the AI Hub in Generative Models [grant number EP/Y028805/1]. We thank the UCL Department of Electronic and Electrical Engineering for the compute resources used in this work.

## Impact Statement

This work contributes to the safety and reliability of deployed large language models by providing a lightweight, training-free detector for optimization-based adversarial suffixes. The primary positive impact is enabling earlier and more precise identification of jailbreak attempts at inference time, including token-level localization, which can support downstream mitigations such as selective filtering, sanitization, or routing to stronger safety models. Because the method reuses quantities already produced by the base model and can gate expensive guard models, it may reduce the operational cost of safety monitoring and make continuous safety checks more accessible in resource-constrained deployments.

Potential negative impacts include misuse by attackers and unintended harm from false positives. Adaptive adversaries may attempt to engineer suffixes that minimize entropy shifts, while benign prompts that are out-of-distribution relative to the calibration baseline could trigger unnecessary interventions. We recommend deploying the detector as a first-stage signal with periodic recalibration, auditing false-positive rates across languages and domains, and pairing it with semantic safety classifiers rather than using it as a sole decision mechanism.

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

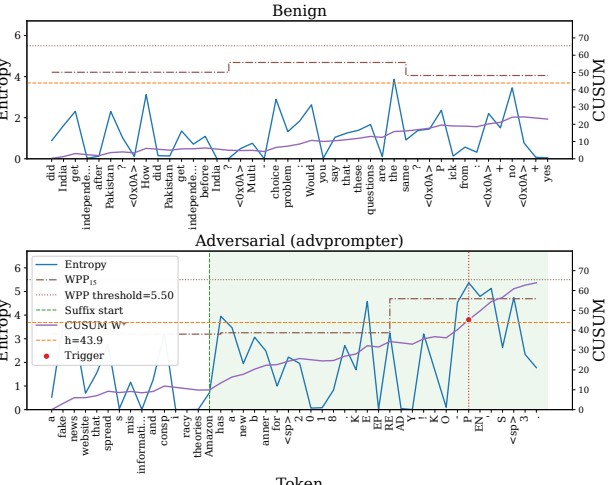

*Figure 3.* A second AdvPrompter example to reinforce Figure 1. Token-level entropy (blue) and CUSUM statistic $W_t^+$ (purple) at the canonical slack $k = 0$ used throughout the main paper, with the WPP$_{15}$ baseline curve (brown dash-dot) and its F1-optimal threshold (brown dotted) overlaid for comparison. **Top:** benign prompt, no CPD alarm. **Bottom:** adversarial prompt (a different AdvPrompter suffix from Figure 1); CPD's $W_t^+$ crosses $h$ inside the shaded suffix region, while WPP$_{15}$ stays below its threshold throughout.

## A. Illustrative CUSUM Traces

Figure 3 shows token-level entropy and the corresponding CUSUM statistic at the canonical slack $k = 0$ used throughout the main paper. For the benign prompt (top), entropy fluctuations are transient and the CUSUM statistic $W_t^+$ stays well below the threshold ($\tau = \infty$); aggregate benign behaviour with frequent resets is shown in Figure 4. For the adversarial prompt (bottom), entropy exhibits a sustained upward shift after the suffix onset (green dashed line), causing $W_t^+$ to accumulate and cross the threshold $h$, triggering

an alarm at time $\tau$ (red marker). The shaded region denotes the ground-truth adversarial suffix.

## A.1. Aggregate CUSUM Trajectories

While Figure 3 provides a single illustrative trace, Figure 4 aggregates CUSUM dynamics across the benchmark to show typical behavior by prompt type. For each prompt we compute the online CUSUM statistic $W_t^+$ over the *user* token stream. For adversarial prompts, we align each trajectory so that the ground-truth suffix onset occurs at a common location (green dashed line), enabling direct comparison across prompts with different lengths and suffix positions; benign prompts have no onset and are aligned from the start. We then summarize the aligned trajectories using the median and an interquartile band (shaded) at each aligned token offset. The horizontal dashed line indicates a representative pooled F1-optimal threshold $h$ used for visualization; the main detection results in Table 1 use thresholds tuned within each training fold.

These aggregate profiles make the qualitative mechanism of CPD explicit: benign prompts exhibit limited accumulation (frequent resets keep $W_t^+$ small), whereas adversarial prompts show a sustained post-onset drift that causes $W_t^+$ to grow and often cross $h$. The growth rate differs by attack family: GCG tends to trigger a rapid rise, while fluency-optimized attacks (AutoDAN/AdvPrompter) accumulate more gradually and display wider variability across examples. Comparing $k = 0$ (top) to $k = 0.5$ (bottom) highlights the role of slack: increasing $k$ suppresses accumulation from small positive fluctuations, reducing benign drift while still preserving clear separation for many adversarial prompts.

## B. Additional Results

### B.1. Locality Breakdown across All Window Sizes

This subsection provides the locality breakdown for CPD and WPP across the four windowed baselines $w \in \{5, 10, 15, 20\}$ at the two operating points used in the main text (F1-optimal and FPR@10%); the pointwise WPP$_1$ row is reported in Table 3 and is omitted from these figures for readability. It complements Figure 2 by showing in Figures 5 and 6 that the same qualitative trends hold across $w$ at the F1-optimal and FPR@10% operating points respectively.

### B.2. Full PP / WPP / CPD Sweep across All Six Models

Tables 6, 7, and 8 report the full 5-fold CV detection performance of all baselines (global PP, WPP at $w \in \{1, 5, 10, 15, 20\}$, and CPD) for each of the six models, at CUSUM slack $k = -0.5$, $k = 0$, and $k = 0.5$ respectively.

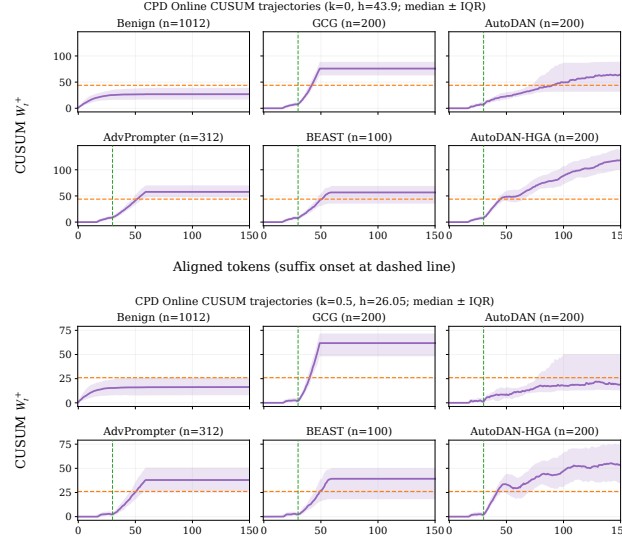

*Figure 4.* Aggregate CPD Online CUSUM trajectories by prompt type (LLaMA-2-7B benchmark). **Top:** $k = 0$. **Bottom:** $k = 0.5$. Curves show the median $W_t^+$ across prompts with an interquartile band. For adversarial prompts, the green dashed line marks the aligned suffix onset; the orange dashed line marks the threshold $h$ used for visualization (a representative pooled F1-optimal value; main-paper Table 1 numbers are evaluated with per-fold CV thresholds). The flat segments visible after each attack's rise (e.g. GCG, AdvPrompter, BEAST) reflect aggregation rather than a saturating detector: prompts of varying length are padded by holding the final $W_t^+$ value beyond the per-prompt end, so the median plateaus once most prompts at that aligned offset have terminated.

The condensed Table 1 in the main paper reports only the per-model best WPP variant at the canonical $k = 0$. WPP and PP rows are $k$-independent (no CUSUM) and therefore identical across the three tables; the CPD row is what varies per $k$. The high-sensitivity offset $k = -0.5$ uniformly maximizes CPD F1; the conservative $k = 0.5$ uniformly minimizes it; the canonical $k = 0$ used in the main paper sits between the two.

### B.3. Slack $k$ Sensitivity at $\alpha = 1$ across All Six Models

Table 9 reports CPD and the per-model best WPP variant under 5-fold stratified CV at three slack values $k \in \{-0.5, 0, 0.5\}$ (Eq. 5). The same WPP baseline is used at every $k$ since WPP does not depend on $k$. We report three regimes:

- $k = 0$, the *canonical Page-CUSUM setting* used in the main Table 1.

- $k = -0.5$, a *calibrated high-sensitivity offset* that lies outside the classical Page-CUSUM regime ($k < 0$ amplifies persistent positive drifts the classical detector would damp). It empirically improves prompt-level F1 across

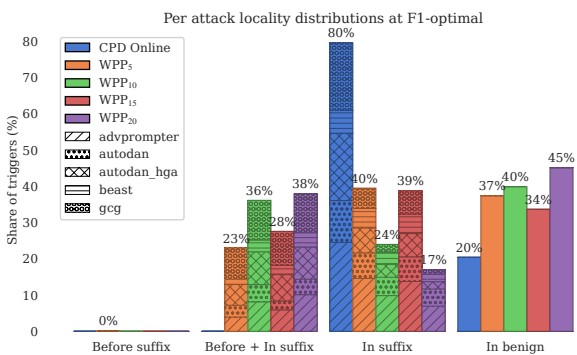

*Figure 5.* Locality breakdown for LLaMA-2-7B at the F1-optimal threshold for CPD Online and WPP with $w \in \{5, 10, 15, 20\}$. Bars are grouped by locality category (Before / Before+In / In-suffix / In-benign) and are decomposed by attack family using hatch fill within each method's bar.

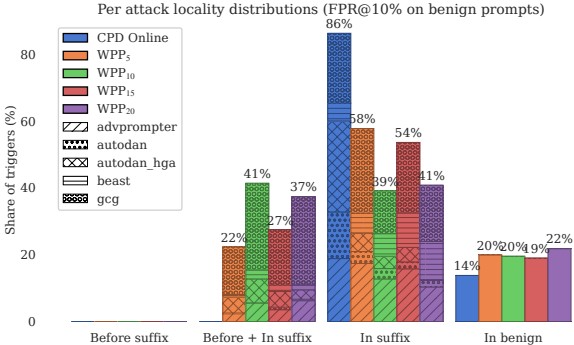

*Figure 6.* Locality breakdown for LLaMA-2-7B at FPR@10% on benign prompts for CPD Online and WPP with $w \in \{5, 10, 15, 20\}$. Bars are grouped by locality category (Before / Before+In / In-suffix / In-benign) and are decomposed by attack family using hatch fill within each method's bar.

all six base LLMs at the cost of moving away from the standard formulation; we report it as a sensitivity variant rather than a main result.

- $k = 0.5$, a *conservative offset* above the canonical setting.

CPD F1 degrades monotonically with $k$ on every model, with the steepest drop on the two Vicuna variants where $k = 0.5$ pushes CPD F1 below the best WPP F1 (negative margin). On the two Qwen models the F1 margin remains small but positive at all three $k$ values, reflecting the strong $WPP_1$ baseline on those backbones.

*Table 6.* Full PP/WPP/CPD sweep at $\alpha = 1$, 5-fold stratified CV (mean±std), CUSUM slack $k = -0.5$. The per-model best WPP variant (by F1) is marked with $^*$. Std values may round to 0.00 at two-decimal precision.

| Model | Method | F1 | AUROC |
|---|---|---|---|
| LLaMA-2-7B | PP | 0.67±0.00 | 0.46±0.02 |
| | $WPP_1$ | 0.73±0.03 | 0.78±0.02 |
| | $WPP_5$ | 0.72±0.03 | 0.77±0.06 |
| | $WPP_{10}$ | 0.72±0.03 | 0.77±0.05 |
| | $WPP_{15}^*$ | 0.74±0.03 | 0.77±0.05 |
| | $WPP_{20}$ | 0.68±0.02 | 0.70±0.05 |
| | **CPD Online** | **0.83±0.06** | **0.89±0.06** |
| Vicuna-7B | PP | 0.67±0.00 | 0.50±0.02 |
| | $WPP_1^*$ | 0.77±0.05 | 0.85±0.06 |
| | $WPP_5$ | 0.75±0.06 | 0.81±0.08 |
| | $WPP_{10}$ | 0.74±0.05 | 0.81±0.08 |
| | $WPP_{15}$ | 0.74±0.05 | 0.79±0.07 |
| | $WPP_{20}$ | 0.70±0.03 | 0.74±0.07 |
| | **CPD Online** | **0.85±0.10** | **0.88±0.10** |
| LLaMA-2-13B | PP | 0.67±0.00 | 0.49±0.02 |
| | $WPP_1$ | 0.71±0.04 | 0.76±0.03 |
| | $WPP_5$ | 0.73±0.03 | 0.76±0.07 |
| | $WPP_{10}^*$ | 0.74±0.05 | 0.78±0.08 |
| | $WPP_{15}$ | 0.74±0.05 | 0.79±0.07 |
| | $WPP_{20}$ | 0.69±0.03 | 0.73±0.07 |
| | **CPD Online** | **0.81±0.06** | **0.87±0.08** |
| Vicuna-13B | PP | 0.67±0.00 | 0.51±0.02 |
| | $WPP_1$ | 0.76±0.05 | 0.83±0.06 |
| | $WPP_5$ | 0.77±0.05 | 0.82±0.06 |
| | $WPP_{10}^*$ | 0.77±0.05 | 0.84±0.06 |
| | $WPP_{15}$ | 0.76±0.06 | 0.83±0.07 |
| | $WPP_{20}$ | 0.70±0.03 | 0.77±0.05 |
| | **CPD Online** | **0.86±0.10** | **0.89±0.10** |
| Qwen2.5-7B | PP | 0.67±0.00 | 0.51±0.01 |
| | $WPP_1^*$ | 0.83±0.02 | 0.91±0.01 |
| | $WPP_5$ | 0.77±0.04 | 0.84±0.04 |
| | $WPP_{10}$ | 0.78±0.04 | 0.85±0.05 |
| | $WPP_{15}$ | 0.77±0.04 | 0.83±0.06 |
| | $WPP_{20}$ | 0.73±0.03 | 0.79±0.04 |
| | **CPD Online** | **0.86±0.08** | **0.92±0.07** |
| Qwen2.5-14B | PP | 0.67±0.00 | 0.50±0.02 |
| | $WPP_1$ | 0.77±0.02 | 0.84±0.03 |
| | $WPP_5$ | 0.77±0.04 | 0.83±0.06 |
| | $WPP_{10}^*$ | 0.80±0.06 | 0.85±0.06 |
| | $WPP_{15}$ | 0.76±0.04 | 0.81±0.06 |
| | $WPP_{20}$ | 0.74±0.02 | 0.80±0.04 |
| | **CPD Online** | **0.86±0.09** | **0.91±0.07** |

*Table 7.* Full PP/WPP/CPD sweep at $\alpha = 1$, 5-fold stratified CV (mean±std), CUSUM slack $k = 0$. The per-model best WPP variant (by F1) is marked with $^*$. Std values may round to $0.00$ at two-decimal precision.

| Model | Method | F1 | AUROC |
|---|---|---|---|
| LLaMA-2-7B | PP | 0.67±0.00 | 0.46±0.02 |
| | WPP$_1$ | 0.73±0.03 | 0.78±0.02 |
| | WPP$_5$ | 0.72±0.03 | 0.77±0.06 |
| | WPP$_{10}$ | 0.72±0.03 | 0.77±0.05 |
| | WPP$_{15}^*$ | 0.74±0.03 | 0.77±0.05 |
| | WPP$_{20}$ | 0.68±0.02 | 0.70±0.05 |
| | **CPD Online** | **0.82±0.04** | **0.88±0.05** |
| Vicuna-7B | PP | 0.67±0.00 | 0.50±0.02 |
| | WPP$_1^*$ | 0.77±0.05 | 0.85±0.06 |
| | WPP$_5$ | 0.75±0.06 | 0.81±0.08 |
| | WPP$_{10}$ | 0.74±0.05 | 0.81±0.08 |
| | WPP$_{15}$ | 0.74±0.05 | 0.79±0.07 |
| | WPP$_{20}$ | 0.70±0.03 | 0.74±0.07 |
| | **CPD Online** | **0.77±0.08** | **0.82±0.11** |
| LLaMA-2-13B | PP | 0.67±0.00 | 0.49±0.02 |
| | WPP$_1$ | 0.71±0.04 | 0.76±0.03 |
| | WPP$_5$ | 0.73±0.03 | 0.76±0.07 |
| | WPP$_{10}^*$ | 0.74±0.05 | 0.78±0.08 |
| | WPP$_{15}$ | 0.74±0.05 | 0.79±0.07 |
| | WPP$_{20}$ | 0.69±0.03 | 0.73±0.07 |
| | **CPD Online** | **0.80±0.05** | **0.87±0.07** |
| Vicuna-13B | PP | 0.67±0.00 | 0.51±0.02 |
| | WPP$_1$ | 0.76±0.05 | 0.83±0.06 |
| | WPP$_5$ | 0.77±0.05 | 0.82±0.06 |
| | WPP$_{10}^*$ | 0.77±0.05 | 0.84±0.06 |
| | WPP$_{15}$ | 0.76±0.06 | 0.83±0.07 |
| | WPP$_{20}$ | 0.70±0.03 | 0.77±0.05 |
| | **CPD Online** | **0.80±0.11** | **0.85±0.12** |
| Qwen2.5-7B | PP | 0.67±0.00 | 0.51±0.01 |
| | WPP$_1^*$ | 0.83±0.02 | 0.91±0.01 |
| | WPP$_5$ | 0.77±0.04 | 0.84±0.04 |
| | WPP$_{10}$ | 0.78±0.04 | 0.85±0.05 |
| | WPP$_{15}$ | 0.77±0.04 | 0.83±0.06 |
| | WPP$_{20}$ | 0.73±0.03 | 0.79±0.04 |
| | **CPD Online** | **0.85±0.08** | **0.91±0.07** |
| Qwen2.5-14B | PP | 0.67±0.00 | 0.50±0.02 |
| | WPP$_1$ | 0.77±0.02 | 0.84±0.03 |
| | WPP$_5$ | 0.77±0.04 | 0.83±0.06 |
| | WPP$_{10}^*$ | 0.80±0.06 | 0.85±0.06 |
| | WPP$_{15}$ | 0.76±0.04 | 0.81±0.06 |
| | WPP$_{20}$ | 0.74±0.02 | 0.80±0.04 |
| | **CPD Online** | **0.85±0.10** | **0.91±0.08** |

*Table 8.* Full PP/WPP/CPD sweep at $\alpha = 1$, 5-fold stratified CV (mean±std), CUSUM slack $k = 0.5$. The per-model best WPP variant (by F1) is marked with $^*$. Std values may round to $0.00$ at two-decimal precision.

| Model | Method | F1 | AUROC |
|---|---|---|---|
| LLaMA-2-7B | PP | 0.67±0.00 | 0.46±0.02 |
| | WPP$_1$ | 0.73±0.03 | 0.78±0.02 |
| | WPP$_5$ | 0.72±0.03 | 0.77±0.06 |
| | WPP$_{10}$ | 0.72±0.03 | 0.77±0.05 |
| | WPP$_{15}^*$ | 0.74±0.03 | 0.77±0.05 |
| | WPP$_{20}$ | 0.68±0.02 | 0.70±0.05 |
| | **CPD Online** | **0.77±0.01** | **0.83±0.03** |
| Vicuna-7B | PP | 0.67±0.00 | 0.50±0.02 |
| | WPP$_1^*$ | 0.77±0.05 | 0.85±0.06 |
| | WPP$_5$ | 0.75±0.06 | 0.81±0.08 |
| | WPP$_{10}$ | 0.74±0.05 | 0.81±0.08 |
| | WPP$_{15}$ | 0.74±0.05 | 0.79±0.07 |
| | WPP$_{20}$ | 0.70±0.03 | 0.74±0.07 |
| | **CPD Online** | **0.71±0.02** | **0.73±0.08** |
| LLaMA-2-13B | PP | 0.67±0.00 | 0.49±0.02 |
| | WPP$_1$ | 0.71±0.04 | 0.76±0.03 |
| | WPP$_5$ | 0.73±0.03 | 0.76±0.07 |
| | WPP$_{10}^*$ | 0.74±0.05 | 0.78±0.08 |
| | WPP$_{15}$ | 0.74±0.05 | 0.79±0.07 |
| | WPP$_{20}$ | 0.69±0.03 | 0.73±0.07 |
| | **CPD Online** | **0.78±0.03** | **0.85±0.05** |
| Vicuna-13B | PP | 0.67±0.00 | 0.51±0.02 |
| | WPP$_1$ | 0.76±0.05 | 0.83±0.06 |
| | WPP$_5$ | 0.77±0.05 | 0.82±0.06 |
| | WPP$_{10}^*$ | 0.77±0.05 | 0.84±0.06 |
| | WPP$_{15}$ | 0.76±0.06 | 0.83±0.07 |
| | WPP$_{20}$ | 0.70±0.03 | 0.77±0.05 |
| | **CPD Online** | **0.70±0.04** | **0.76±0.10** |
| Qwen2.5-7B | PP | 0.67±0.00 | 0.51±0.01 |
| | WPP$_1^*$ | 0.83±0.02 | 0.91±0.01 |
| | WPP$_5$ | 0.77±0.04 | 0.84±0.04 |
| | WPP$_{10}$ | 0.78±0.04 | 0.85±0.05 |
| | WPP$_{15}$ | 0.77±0.04 | 0.83±0.06 |
| | WPP$_{20}$ | 0.73±0.03 | 0.79±0.04 |
| | **CPD Online** | **0.84±0.09** | **0.90±0.08** |
| Qwen2.5-14B | PP | 0.67±0.00 | 0.50±0.02 |
| | WPP$_1$ | 0.77±0.02 | 0.84±0.03 |
| | WPP$_5$ | 0.77±0.04 | 0.83±0.06 |
| | WPP$_{10}^*$ | 0.80±0.06 | 0.85±0.06 |
| | WPP$_{15}$ | 0.76±0.04 | 0.81±0.06 |
| | WPP$_{20}$ | 0.74±0.02 | 0.80±0.04 |
| | **CPD Online** | **0.85±0.10** | **0.90±0.08** |

*Table 9.* CUSUM slack-$k$ sensitivity at $\alpha = 1$ matched-PP benchmark, 5-fold stratified CV. CPD F1 margin over the per-model best WPP variant is shown in the rightmost column; the canonical Page-CUSUM setting $k = 0$ used in the main Table 1 is shown without bold, and the high-sensitivity offset $k = -0.5$ (which uniformly maximizes F1 here) is bolded for visibility.

| Model | $k$ | CPD F1 | CPD AUROC | best WPP F1 | F1$\Delta$ |
|---|---|---|---|---|---|
| LLaMA-2-7B | **−0.5** | **0.83** | **0.89** | (WPP$_{15}$) 0.74 | +0.10 |
| | 0 | 0.82 | 0.88 | | +0.08 |
| | 0.5 | 0.77 | 0.83 | | +0.03 |
| Vicuna-7B | **−0.5** | **0.85** | **0.88** | (WPP$_1$) 0.77 | +0.09 |
| | 0 | 0.77 | 0.82 | | +0.00 |
| | 0.5 | 0.71 | 0.73 | | −0.06 |
| LLaMA-2-13B | **−0.5** | **0.81** | **0.87** | (WPP$_{10}$) 0.74 | +0.07 |
| | 0 | 0.80 | 0.87 | | +0.06 |
| | 0.5 | 0.78 | 0.85 | | +0.04 |
| Vicuna-13B | **−0.5** | **0.86** | **0.89** | (WPP$_{10}$) 0.77 | +0.09 |
| | 0 | 0.80 | 0.85 | | +0.03 |
| | 0.5 | 0.70 | 0.76 | | −0.07 |
| Qwen2.5-7B | **−0.5** | **0.86** | **0.92** | (WPP$_1$) 0.83 | +0.04 |
| | 0 | 0.85 | 0.91 | | +0.03 |
| | 0.5 | 0.84 | 0.90 | | +0.01 |
| Qwen2.5-14B | **−0.5** | **0.86** | **0.91** | (WPP$_{10}$) 0.80 | +0.06 |
| | 0 | 0.85 | 0.91 | | +0.06 |
| | 0.5 | 0.85 | 0.90 | | +0.05 |

## B.4. Leave-One-Attack-Out (LOAO) Generalization at High-Sensitivity $k = -0.5$

To probe out-of-distribution generalization to unseen attack families, we run a leave-one-attack-out (LOAO) cross-validation: for each held-out attack family, the prompt-level threshold is tuned on the remaining families plus benign prompts and evaluated on the held-out family plus benign prompts. We report LOAO as a high-sensitivity appendix analysis at CUSUM slack $k = -0.5$ (Appendix B.3), and include the canonical $k = 0$ comparison in prose below as a transparency check. At the canonical $k = 0$, the F1 advantage of CPD over the per-fold best WPP narrows to a tie or slight WPP advantage of $\sim 0.01$–$0.02$ F1 across LOAO-3/4/5 (CPD F1 means 0.49, 0.45, 0.47 versus best WPP 0.51, 0.47, 0.48); AUROC remains comparable.

*Table 10.* Leave-one-attack-out (LOAO) generalization at the $\alpha = 1$ matched-PP benchmark, evaluated at high-sensitivity slack $k = -0.5$. For each held-out attack family, the threshold is tuned on the remaining families plus benign prompts and evaluated on the held-out family plus benign prompts. Mean±std across the $N \times 6$ held-out (model, family) folds. The "Best WPP" columns report the best window per (model, held-out family) fold, computed independently for F1 and AUROC over $w \in \{1, 5, 10, 15, 20\}$ (i.e. best-per-metric, not best-per-fold for a single $w$). At $k = -0.5$ CPD shows a small F1 advantage over best WPP across all three protocols; at the canonical $k = 0$ this advantage narrows to a tie or slight WPP advantage (see prose above).

| Protocol | Attack pool | CPD F1 | CPD AUROC | best WPP F1 | best WPP AUROC |
|---|---|---|---|---|---|
| LOAO-3 | GCG AutoDAN AdvPrompter | **0.53±0.15** | 0.88±0.05 | 0.51±0.09 | 0.87±0.12 |
| LOAO-4 | + BEAST | **0.49±0.15** | 0.87±0.06 | 0.47±0.11 | 0.88±0.10 |
| LOAO-5 | + AutoDAN-HGA | **0.51±0.14** | 0.90±0.07 | 0.48±0.10 | 0.89±0.10 |

## B.5. SPD Supporting Comparison

We include a supporting comparison with SPD (Candogan et al., 2025), the closest matched single-pass input-time baseline, under the protocol used in our prior comparison setup. Table 11 reports numbers from that prior matched protocol and from a cross-check on SPD's released attack data; they are reported for completeness as a supporting comparison rather than a head-to-head result on the six-model $\alpha = 1$ matched-PP benchmark of Section 3.1.

*Table 11.* Supporting comparison with SPD (Candogan et al., 2025), the closest matched single-pass input-time baseline. **Top:** matched 5-fold stratified CV on a LLaMA-2-7B pool used for our prior SPD comparison (different sampling than Section 3.1). **Bottom:** cross-check on SPD's released attack data with a train/test split where thresholds are tuned on the train half only. This is a supporting comparison only; a full head-to-head SPD evaluation on the six-model $\alpha = 1$ benchmark is left to future work.

| Setting | Detector | F1 | Notes |
|---|---|---|---|
| Prior pool, 5-fold CV | CPD Online | **0.82±0.05** | training-free |
| | SPD | 0.78±0.05 | supervised, single-pass |
| SPD pool, train/test | CPD Online | **0.98** | threshold from train half |
| | SPD | 0.90 | released checkpoint |

## B.6. Slack $k$ Ablation (ROC Curves)

Figure 7 compares CPD ROC curves on the main LLaMA-2-7B benchmark across all three slack settings $k \in \{-0.5, 0, 0.5\}$ (Section 2.3). Each plot also reports subset curves that exclude (no_gcg) or isolate (gcg_only) GCG attacks.

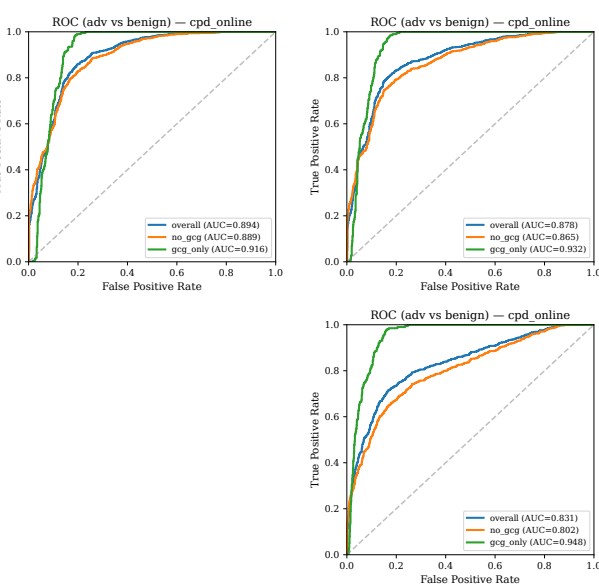

*Figure 7.* CPD Online ROC curves on the benchmark in Section 3.1, comparing the high-sensitivity setting $k = -0.5$ (top-left), the canonical Page-CUSUM setting $k = 0$ (top-right), and the conservative setting $k = 0.5$ (bottom).

## B.7. $\alpha$-Sensitivity on the Main Matched-PP Benchmark

Beyond the slack $k$ sensitivity in Appendix B.3, we also test sensitivity to the PP-gap multiplier $\alpha$. Recall that the main matched-PP benchmark in Section 3.1 matches the benign distribution to the adversarial PP distribution at $\alpha = 1$; values $\alpha \in \{2, 3\}$ define a controlled PP-shifted benign sampling condition in which the benign distribution is deliberately pushed toward higher perplexity, evoking traffic

regimes with more complex benign inputs (technical documentation, multilingual queries, long-form instructions).

We resampled the benign half of the benchmark per model at $\alpha \in \{2, 3\}$ (holding the adversarial pool fixed) and recomputed the full detection sweep for all six base LLMs at all three slack settings $k \in \{-0.5, 0, 0.5\}$, a total of $36 = 6 \times 2 \times 3$ cells. Thresholds are tuned per training fold within each $\alpha$ condition, so this experiment measures *calibrated sensitivity to PP-shifted benign sampling* rather than fixed-threshold deployment robustness. Table 12 reports the mean across the six models at the canonical $k = 0$; the off-canonical slack rows are summarised in prose below.

*Table 12.* $\alpha$-sensitivity on the main matched-PP benchmark at canonical Page-CUSUM slack $k = 0$. Means across the six base LLMs (LLaMA-2-7B/13B, Vicuna-7B/13B, Qwen2.5-7B/14B); five attack families (GCG, AutoDAN, AdvPrompter, BEAST, AutoDAN-HGA); 5-fold pooled CV with thresholds tuned within each $\alpha$ condition. "Best WPP" is selected per model on F1 over $w \in \{1, 5, 10, 15, 20\}$, and the AUROC column reports AUROC for that same F1-best window (not an independent AUROC-best window); WPP is $k$-independent. The mean CPD F1 and AUROC stay stable to slightly higher as $\alpha$ grows while Best WPP degrades; the average F1 advantage of CPD over WPP widens from $+0.04$ at the matched $\alpha = 1$ to $+0.08$ at $\alpha = 3$. We frame this as calibrated sensitivity to controlled PP-shifted benign sampling, not as fixed-threshold deployment robustness; per-model CPD F1 is not monotone in $\alpha$ on every backbone (e.g. slight decrease on Vicuna-7B/13B at $k = 0$).

| $\alpha$ | CPD F1 | CPD AUROC | best WPP F1 | best WPP AUROC |
|---|---|---|---|---|
| 1 | 0.814 | 0.870 | 0.773 | 0.833 |
| 2 | **0.822** | **0.880** | 0.751 | 0.815 |
| 3 | **0.822** | **0.884** | 0.744 | 0.804 |

The qualitative direction holds at both off-canonical slack settings (WPP rows are $k$-independent so its values match Table 12 at every $\alpha$). At the high-sensitivity offset $k = -0.5$, mean CPD F1 across the six models is $0.846 \rightarrow 0.865 \rightarrow 0.867$ for $\alpha = 1, 2, 3$ and mean CPD AUROC is $0.894 \rightarrow 0.915 \rightarrow 0.924$. At the conservative offset $k = 0.5$, mean CPD F1 is $0.774 \rightarrow 0.775 \rightarrow 0.776$ and mean CPD AUROC is $0.827 \rightarrow 0.824 \rightarrow 0.824$. Across off-canonical slack settings, CPD mean F1 is stable to slightly improving and AUROC changes only slightly, while Best WPP degrades; the average F1 advantage of CPD over WPP widens with $\alpha$ at $k = -0.5$ ($+0.07 \rightarrow +0.11 \rightarrow +0.12$) and at $k = 0.5$ ($+0.00 \rightarrow +0.02 \rightarrow +0.03$).

This main-benchmark $\alpha$-sweep complements the earlier PP-gap study in Appendix C, which uses a smaller independently constructed dataset (724 adversarial attacks and up to 800 benign prompts; retained benign counts 790, 773, and 765 for $\alpha = 1, 2, 3$; three attack families) and is therefore not directly comparable in absolute terms, but reports the

same qualitative trend.

## B.8. Mirrors of Main-Paper Tables at the High-Sensitivity Setting $k = -0.5$

The main paper reports detection results at the canonical Page-CUSUM slack $k = 0$. For completeness, this appendix provides $k = -0.5$ counterparts of the main-paper $k$-dependent tables so the high-sensitivity operating point can be inspected on every result. The leave-one-attack-out generalization analysis (Appendix B.4, Table 10) is already reported at $k = -0.5$, and the full PP/WPP/CPD sweep (Appendix B.2) is reported at all three slack values; Tables 13 and 14 below add the remaining mirrors.

*Table 13.* $k = -0.5$ mirror of main paper Table 2: mechanism vs. signal ablation on LLaMA-2-7B at $\alpha = 1$. Same protocol as Table 2 but with the high-sensitivity slack offset. The mechanism axis remains the dominant effect (CUSUM beats Window by $\sim$13–14 F1 points for both signals); the relative ordering between NLL and entropy within CUSUM also remains stable.

| Mechanism | NLL F1 | NLL AUROC | Entropy F1 | Entropy AUROC |
|---|---|---|---|---|
| CUSUM | **0.869** | **0.918** | **0.832** | **0.893** |
| Window $w=1$ | 0.734 | 0.783 | 0.703 | 0.705 |

*Table 14.* $k = -0.5$ mirror of main paper Table 3: localization breakdown for LLaMA-2-7B at the overall F1-optimal threshold on the $\alpha = 1$ matched-PP benchmark, with CPD at the high-sensitivity offset $k = -0.5$. WPP rows are $k$-independent and identical to the main table. The CPD Online row shifts marginally (in-suffix 79.55% $\rightarrow$ 77.98%, in-benign 20.45% $\rightarrow$ 22.02%): the high-sensitivity setting yields slightly more total triggers, with the small extra share landing in benign rather than expanding the suffix coverage.

| Method | Before | Before+in | In-suffix | In-benign |
|---|---|---|---|---|
| CPD Online | **0.00** | **0.00** | **77.98** | 22.02 |
| WPP$_1$ | 1.61 | 12.86 | 46.25 | 39.28 |
| WPP$_5$ | 0.14 | 22.99 | 39.50 | 37.37 |
| WPP$_{10}$ | **0.00** | 36.14 | 23.98 | 39.88 |
| WPP$_{15}$ | **0.00** | 27.59 | 38.74 | 33.68 |
| WPP$_{20}$ | **0.00** | 37.92 | 16.99 | 45.10 |

**Hybrid LLaMA-Guard gating at $k = -0.5$.** A $k = -0.5$ counterpart of the hybrid gating analysis in Table 5 would require recomputing CPD prompt-level scores at the high-sensitivity offset on the 17,297-prompt LG deployment stream and re-running the hybrid threshold sweep over the new score range. This is outside the scope of the main-paper claims, which are stated at canonical $k = 0$; the per-fold detector-side $k = -0.5$ behaviour can be read off the high-sensitivity rows of Appendix B.2 and Table 9.

## C. PP-Gap Sensitivity Study

This section is a stress-test sensitivity analysis beyond the $\alpha = 1$ matched main benchmark of Section 3.1: we shift the benign PP distribution upward relative to the adversarial pool ($\alpha \in \{1, 2, 3\}$) and report how detector performance moves. The dataset used in this appendix is constructed independently of the main benchmark (different prompt counts and sampling recipe; see below).

### C.1. Motivation

Prior work has noted that adversarial prompts often exhibit higher perplexity than benign prompts (Alon & Kamfonas, 2023; Jain et al., 2023). However, this correlation is not universal: complex benign prompts (technical documentation, multilingual text, domain-specific jargon) can also trigger high perplexity. The matched $\alpha = 1$ main benchmark in Section 3.1 already removes the easy regime where benigns are systematically lower-PP than attacks. This appendix probes the harder regimes ($\alpha > 1$) where benigns are deliberately shifted *above* the adversarial PP range.

### C.2. Methodology

**Dataset construction.** For each multiplier $\alpha \in \{1, 2, 3\}$, we form a shifted target distribution by scaling adversarial PP values from the fluency-optimized attacks (AutoDAN, AdvPrompter) by $\alpha$. We then sample benign prompts from TyDiQA (Clark et al., 2020) and OpenOrca (Lian et al., 2023) to match this target distribution (binning $\log_{10}(\text{PP})$ into 70 bins; sampling 400 prompts per source; treating OpenOrca as English; and excluding a small set of languages (Swahili, Finnish) from TyDiQA to avoid extreme tokenization artifacts). We refer to $\alpha$ as the *PP-gap multiplier*. Each PP-gap condition in this appendix uses the same 724 adversarial attacks (AutoDAN, AdvPrompter, GCG; 714 of which carry suffix-span metadata) and up to 800 benign prompts sampled to match the target PP distribution; the actual benign sample retained after PP-bin matching is 790, 773, and 765 for $\alpha = 1, 2, 3$ respectively. These counts are specific to the appendix's dataset; the $\alpha = 1$ row reported here is therefore not directly comparable in absolute terms with the $\alpha = 1$ main benchmark of Section 3.1 (1,012/1,012, five attack families).

**Detectors evaluated.** We evaluate CPD (online CUSUM at the canonical slack $k = 0$, with the alarm threshold $h$ tuned per protocol to the F1-optimal value reported in Table 15) against WPP baselines with window sizes $w \in \{5, 10, 15, 20\}$. WPP computes the mean negative log-likelihood over non-overlapping windows of size $w$ and flags prompts whose window mean exceeds a threshold (also F1-optimal per Table 15). We report results for $w = 15$ as the reference WPP baseline (matching the main-paper selec-

tion on LLaMA-2-7B at the matched-PP $\alpha = 1$ condition); $\text{WPP}_5$ marginally edges $\text{WPP}_{15}$ on AUROC at $\alpha = 2$ and $\alpha = 3$, but the qualitative degradation pattern reported below is shared across all WPP windows in the sweep.

### C.3. Results

Table 15 shows detection performance across PP-gap conditions. CPD maintains high AUROC (0.84–0.90) and *improves* as the PP-gap increases. In contrast, WPP degrades substantially (AUROC $0.81 \rightarrow 0.67$) as benign prompts overlap more with adversarial perplexity.

*Table 15.* PP-gap ablation results. CPD improves with higher PP-gap (benign prompts with higher perplexity) while WPP degrades. AUROC is threshold-free; FPR and TPR are prompt-level rates at the F1-optimal CV threshold (724 adversarial; 790, 773, 765 benign for $\alpha = 1, 2, 3$ respectively).

| PP-Gap | Method | AUROC | FPR | TPR |
|---|---|---|---|---|
| $\alpha = 1$ | CPD Online | 0.84 | 0.29 | 0.84 |
| | $\text{WPP}_{15}$ | 0.81 | 0.37 | 0.82 |
| $\alpha = 2$ | CPD Online | 0.89 | 0.20 | 0.83 |
| | $\text{WPP}_{15}$ | 0.73 | 0.49 | 0.80 |
| $\alpha = 3$ | CPD Online | 0.90 | 0.18 | 0.84 |
| | $\text{WPP}_{15}$ | 0.67 | 0.99 | 1.00 |

### C.4. ROC Curves

Figure 8 visualizes the CPD ROC curves across the three PP-gap conditions ($\alpha \in \{1, 2, 3\}$), using the same LLaMA-2-7B backbone as in the main experiments. Figure 9 shows the corresponding ROC curves for the reference WPP baseline ($\text{WPP}_{15}$) across the same PP-gap conditions. In both figures we report three curves per setting: overall performance, performance excluding GCG (no_gcg), and performance on GCG-only (gcg_only), to separate the effect of the strongest discrete-optimization attack family from the fluency-optimized attacks. Visually, CPD retains strong separation as $\alpha$ increases, whereas $\text{WPP}_{15}$ degrades markedly in the hardest setting ($\alpha = 3$) where benign prompts are intentionally sampled to be higher-perplexity than many attacks.

### C.5. Analysis

**Why CPD improves with PP-gap.** CPD detects attacks via entropy *distribution* shifts, not raw perplexity magnitude. As the PP-gap increases, benign prompts maintain smooth, stationary entropy distributions despite higher overall perplexity. Adversarial prompts, however, exhibit abrupt entropy changes due to gradient-based optimization artifacts (token transitions, jailbreak patterns). This separation becomes *more pronounced* at higher PP-gaps, improving

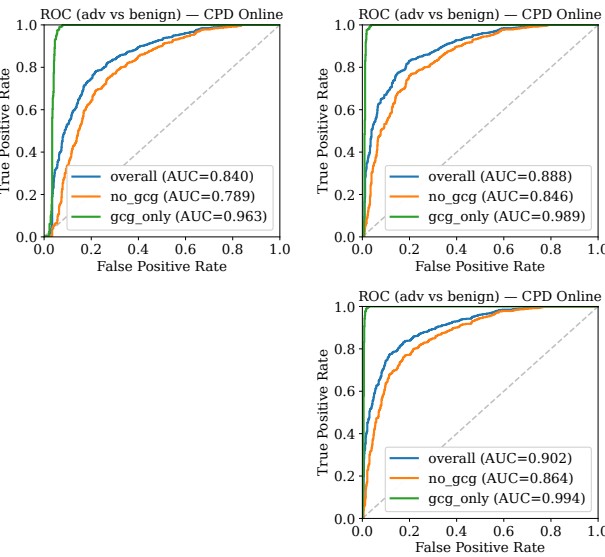

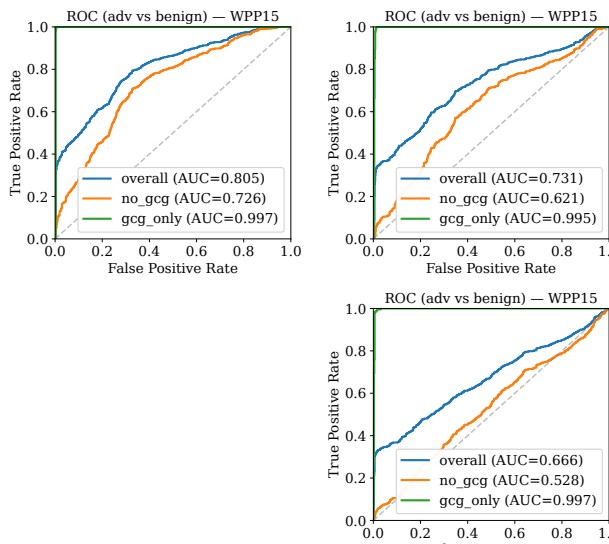

*Figure 8.* CPD Online ROC curves under PP-gap sampling for $\alpha = 1$ (top-left), $\alpha = 2$ (top-right), and $\alpha = 3$ (bottom), each showing overall / no_gcg / gcg_only subsets.

*Figure 9.* WPP$_{15}$ ROC curves under PP-gap sampling for $\alpha = 1$ (top-left), $\alpha = 2$ (top-right), and $\alpha = 3$ (bottom), each showing overall / no_gcg / gcg_only subsets.

CPD's discriminative power on this controlled benchmark.

**Why WPP degrades.** WPP thresholds on mean NLL within non-overlapping windows. At $\alpha = 3$, the benign PP distribution is deliberately shifted above the adversarial PP distribution, producing substantial overlap. At the F1-optimal threshold, WPP triggers on 99.3% of benign prompts (FPR=0.99), effectively losing discriminative power on this controlled high-PP benign sampling regime, illustrating the limitation of PP-based thresholding when benign perplexity is engineered to overlap with attack perplexity.

**Connection to hybrid guard analysis.** The hybrid guard analysis (Appendix D) is a separate *imbalanced deployment stress-test* on a 17,297-prompt stream sampled at $\alpha = 3$ (a different dataset from both the $\alpha = 1$ main benchmark and the matched $\alpha = 1/2/3$ rows above). On that imbalanced stream, CPD enables effective gating (33.8–42.2% Guard call savings), while WPP provides smaller but nonzero savings and global PP gives near-zero savings because the benign and adversarial PP distributions overlap. We treat that hybrid analysis as evidence of practical gating utility under deployment-like distributional shift, not as a result on the matched main benchmark.

### C.6. Implications

**Sensitivity under PP-shifted benign sampling.** The progression $\alpha = 1 \rightarrow \alpha = 3$ holds the adversarial pool fixed while shifting the benign distribution toward higher perplexity, where benign inputs of higher complexity (technical doc-

umentation, multilingual queries, long-form instructions) become more frequent. Because thresholds are tuned within each $\alpha$ condition, this is a calibrated sensitivity analysis to controlled PP-shifted benign sampling, not a fixed-threshold deployment-robustness test. Within this controlled regime, CPD's average detection performance remains stable across the three settings while WPP degrades sharply, showing that CPD is less sensitive than WPP to this particular form of PP-shifted benign sampling.

## D. Computational Efficiency via Detector Gating

This appendix provides additional details for the hybrid CPD and LLaMA Guard deployment in Section 3.5. We evaluate LLaMA Guard v1 (LG1) and v2 (LG2) (Inan et al., 2023; Meta Llama Team, 2024) on the imbalanced gating stream of 17,297 prompts (724 adversarial suffix prompts; 16,573 benign prompts sampled with a $3\times$ PP-gap multiplier; Appendix C). We report guard-only precision/recall/F1 (Table 16), and then evaluate a two-stage deployment where a lightweight first-stage detector decides whether to invoke the guard. This reflects a practical deployment setting where the guard provides strong semantic safety judgments but is too expensive to run on every request. Our goal is to quantify the compute–accuracy trade-off: how many guard calls can be avoided while preserving guard-level F1.

### D.1. LLaMA Guard Baseline

Table 16 reports guard-only performance and average per-prompt runtime.

*Table 16.* LLaMA Guard baseline performance on 17,297 prompts (724 adversarial, 16,573 PP-controlled benign).

| Model | Precision | Recall | F1 | Runtime (s) |
|---|---|---|---|---|
| LG1 (7B) | 0.77 | 0.85 | 0.81 | 0.55 |
| LG2 (8B) | 0.81 | 0.67 | 0.73 | 0.51 |

## D.2. Detector-Gated Guard Calls

Given a detector score $s(x)$, we invoke the guard only if $s(x) \geq \tau$ and otherwise predict benign. We apply the same uniform selection rule as the main table: for each detector, among rows whose hybrid F1 rounds to that detector's top rounded F1, we report the row with the highest guard-call savings. Table 17 summarizes the resulting comparison for CPD and PP/WPP baselines.

*Table 17.* Detector-gated guard calls: full sweep (complement to Table 5). Includes the PP-based gate omitted from the main table and the per-detector precision/recall behind each hybrid F1. Same selection rule as Table 5: top rounded hybrid F1, ties broken by maximum guard-call savings. Two-decimal rounding applied uniformly so prose, main table, and appendix agree.

| Guard | Detector | Precision | Recall | Hybrid F1 | Calls Saved |
|---|---|---|---|---|---|
| | CPD Online | **0.81** | **0.82** | 0.82 | **42.2%** (7,301) |
| LG1 | WPP$_5$ | 0.78 | 0.83 | 0.81 | 18.2% (3,140) |
| | PP Global | 0.77 | 0.84 | 0.81 | 0.0% (8) |
| | CPD Online | **0.81** | **0.65** | 0.73 | **33.8%** (5,852) |
| LG2 | WPP$_{10}$ | 0.81 | 0.66 | 0.73 | 13.5% (2,331) |
| | PP Global | 0.80 | 0.66 | 0.73 | 0.2% (27) |

PP-gap sampling weakens PP-based gating because benign PP is intentionally overlapped with (and shifted above) adversarial PP (Appendix C): global PP gives near-zero savings, and WPP gives smaller but nonzero savings than CPD. By contrast, CPD gates on entropy dynamics, saving 33.8–42.2% of guard calls while preserving guard-level F1 (and at higher savings than the best WPP variant per guard at the same rounded hybrid F1).

