# OpenReview forum: "Detecting Fluent Optimization-Based Adversarial Prompts via Sequential Entropy Changes"
_ICML.cc/2026/Conference — ICML 2026 regular_

### Official Review · Reviewer_gQMi · 2026-03-09

**Soundness:** 1
**Presentation:** 2
**Significance:** 1
**Originality:** 1
**Overall Recommendation:** 2
**Confidence:** 4

**Summary:**

This work proposes casting adversarial suffix detection as an online change-point detection problem, where token entropies are treated as a sequence and normalized with a CUSUM statistic. When the CUSUM statistic goes avobe a pre-specified threshold, the start of an adversarial suffix is triggered and the user prompt is flagged as adversarial. Authors evaluate their method on Llama2 and Vicuna models against windowed-perplexity-based detectors and the more expensive safety classifiers Llama Guard v1 and v2. Authors show an improved F1 and AUROC when compared against perplexity-based detectors and a reduction of expensive Llama Guard calls when combined as a mixed approach.

**Compliance With Llm Reviewing Policy:**

Affirmed.

**Key Questions For Authors:**

- In Algorithm 1, how can you trigger multiple times for a given prompt if whenever $W>h$ you return and finish the execution?

- What is the motivation behind multiplying adversarial PP by 3? Is this valid?

- In lines 233-234 left column you state that you perform 5-fold stratified cross validation. Do you reuse the same 5 training/validation splits for all methods?

- Have you tried $w=1$? Why do you start at $w=5$?

**Limitations:**

Authors discuss the limitations of their method.

**Strengths And Weaknesses:**

## Strenghts

- The paper is well written and the method is quite simple and straightfoward to understand. Figure 1 helps understand the core idea of the approach.

- The improvements over windowed perplexity detectors is clear in terms of F1, AUROC and Llama Guard Calls saved for the studied window sizes and experimental setup.

## Weaknesses

- **Limited novelty.** The proposed method is an application of the existing CUMSUM technique. While change-point detection methods are credited in lines 423 left column - 389 right column, The connection to the existing CUMSUM literature is unclear. A proper discussion of the design choices in the method and the relationship with the literature is needed.

- **Unclear source of improvements and non-standard experimental setup.** The key argument you use to employ CUSUM is that attacks expicitly optimize obtaining a low global perplexity. However, you don't evaluate a CUSUM-based approach using per-token perplexity or a windowed-perplexity approach with $w=1$. Without these experiments it is unclear weather your improvements come from using per-token statistics or from using entropy instead of perplexity. Additionally, in lines 213-217 right column, you state that adversarial PP values are multiplied by $\alpha = 3$. This seems odd and I couln't understand the reasons behind this, specially when in appendix C, perplexity-based methods perform much worse with larger $\alpha$. Another point is that the evaluated models (Llama2 and Vicuna) are close to 3 years old. It would be nice to see how the proposed defense performs with newer models.

- **Key references and baselines missing.** While SmoothLLM is cited, there is no evaluation against it. Similarly, other methods like RA-LLM [1], Self-Defense [2] or SPD [3] are not cited or compared against.

## References

[1] Cao et al., Defending against alignment-breaking attacks via robustly aligned llm. arXiv preprint arXiv:2309.14348 2023.

[2] Phute et al., LLM Self Defense: By Self Examination, LLMs Know They Are Being Tricked. ICLR Tiny Paper 2024.

[3] Candogan et al., Single-pass Detection of Jailbreaking Input in Large Language Models. TMLR 2025.

---

> ### Author Rebuttal · Authors · 2026-03-30
>
> Thank you for the thorough feedback. We ran new experiments during the rebuttal period to address each point.
>
> 1) On the source of gains: we ran a 2×2 ablation crossing CUSUM vs windowed thresholding with NLL vs entropy. We also added w=1 (per-token NLL), which turns out to be the strongest windowed baseline overall. Results at `k=0` under 5-fold CV:
>
> | Mechanism | NLL (F1/AUROC) | Entropy (F1/AUROC) |
> |-----------|---------------|-------------------|
> | CUSUM | 0.89 / 0.94 | 0.82 / 0.91 |
> | Window w=1 | 0.72 / 0.75 | 0.69 / 0.70 |
>
> The CUSUM-vs-window gap is 13–17 F1 points regardless of signal, while the NLL-vs-entropy gap under the same mechanism is only 3–7 F1 points. The improvement comes from sequential accumulation, not the particular token statistic.
>
> On the connection to CUSUM literature: as we discuss in our limitations (Section 5), our setting departs from classical sequential change detection in important ways. Token entropies are non-IID with strong serial dependence, we apply CPD within a single prompt rather than across repeated samples, and the system-prompt baseline via median/MAD replaces external calibration data. We do not claim the optimality guarantees of classical theory. CUSUM works well here as a heuristic despite the non-standard setting. We will expand this discussion in the final version.
>
> 2) We extended Table 1 by adding BEAST (Sadasivan et al., 2024), a beam-search fluency attack, giving 4 attack families across all 6 models:
>
> | Model | CPD F1/AUROC | Best WPP F1/AUROC |
> |-------|-------------|------------------|
> | LLaMA-2-7B | 0.82 / 0.90 | 0.74 / 0.74 (w=1) |
> | LLaMA-2-13B | 0.82 / 0.90 | 0.71 / 0.73 (w=5) |
> | Vicuna-7B | 0.76 / 0.83 | 0.76 / 0.83 (w=1) |
> | Vicuna-13B | 0.81 / 0.86 | 0.76 / 0.83 (w=1) |
> | Qwen-7B | 0.87 / 0.93 | 0.81 / 0.88 (w=1) |
> | Qwen-14B | 0.87 / 0.91 | 0.74 / 0.75 (w=1) |
>
> CPD outperforms the best windowed baseline on F1 for 5 of 6 models at `k=0`. On Vicuna-7B the two are effectively tied. The Qwen results show the method generalizes to newer architectures.
>
> 3) On the PP-gap multiplier: in real deployments, benign prompts span a wide range of perplexity — technical documentation, multilingual text, and domain jargon all produce high PP. The α=3 multiplier deliberately shifts the benign PP distribution *above* the adversarial PP distribution to stress-test detectors in this realistic regime. As shown in our PP-gap ablation (Table 5, Appendix C), CPD actually improves with higher α (AUROC 0.84→0.90) while the best WPP degrades (0.81→0.72), because CPD detects distributional shifts rather than absolute perplexity levels.
>
> 4) On protocol and window sizes (Q3, Q4): yes, we use the same 5-fold stratified splits across all methods. Our original w∈{5,10,15,20} followed Jain et al. (2023, "Baseline defenses for adversarial attacks"), who tested w∈{2,5,10,15,20} and recommended w=10 for stability. After this feedback we added w=1 and found it is the strongest windowed baseline overall — but CPD still outperforms it by 13–17 F1 points. The advantage is the cumulative evidence mechanism, not window granularity.
>
> 5) On Algorithm 1: it returns on the first alarm (line 12 of the algorithm) for the prompt-level detection decision. The alarm index provides the localization estimate. In our locality evaluation (Section 3.4) we analyze the trigger position distributions separately.
>
> 6) On baselines: we report SPD (Candogan et al., 2025) as the closest matched baseline in our single-pass input-detection setting: CPD F1=0.82±0.05 vs SPD F1=0.78±0.05 under identical 5-fold CV. As an external check on SPD's released attack data (train/test split; thresholds tuned on train only), CPD F1=0.98 vs SPD 0.90. SPD requires supervised training and partial response generation; CPD is training-free, operates at input time, and provides token-level localization. SmoothLLM/RA-LLM (multi-pass perturbation) and Self-Defense (post-generation) detect harmful *output* rather than adversarial *input*; we will report matched-protocol results for them under a LLaMA Guard judge in revision.
>
> The 2×2 ablation, w=1 results, 6-model extended Table 1, and SPD comparison directly address the concerns raised above.

---

> > ### Author Rebuttal · Reviewer_gQMi · 2026-04-03
> >
> > Dear authors,
> >
> > Thanks for the added experiments. Specially the $w=1$ experiments.
> >
> > I'm still not convinced about the $\alpha$ multiplier. 1. Instead of multiplying the perplexity on your benign dataset, can't you select a more realistic benign dataset? 2. It seems this is not done to make the setup more realistic, but to improve the gap between CPD and perplexity filters. I would advise finding a more realistic benign set or not modifying benign perplexities at all.  Could you repeat the 2x2 ablation with $\alpha=1$?
> >
> > Best regards,
> >
> > Reviewer gQMi

---

> > > ### Author Response · Authors · 2026-04-07
> > >
> > > the perplexity multiplier was a stress test for perplexity based methods. To clarify, we did **not** modify or multiply the perplexity of individual benign prompts; we re-sampled benign prompts to match a target perplexity distribution and study a harder regime for these filters.
> > >
> > > We reran the 2×2 ablation with **alpha=1**
> > > | Mechanism | NLL (F1 / AUROC) | Entropy (F1 / AUROC) |
> > > |---|---|---|
> > > | CUSUM | 0.831 / 0.862 | 0.772 / 0.842 |
> > > | Window w=1 | 0.690 / 0.736 | 0.676 / 0.695 |
> > > All window sizes at alpha=1:
> > > - NLL windows w∈{1,5,10,15,20}: F1 = 0.690, 0.713, 0.703, 0.735, 0.685 (best w=15).
> > > - Entropy windows w∈{1,5,10,15,20}: F1 = 0.676, 0.675, 0.670, 0.672, 0.657 (best w=1).
> > >
> > > CUSUM still outperforms the best windowed baseline on both signals, with a smaller margin (~9.6 F1 points) than in alpha>1 stress tests. We will present alpha=1 as primary and alpha>1 only as sensitivity analysis.

---

### Official Review · Reviewer_iGSq · 2026-03-10

**Soundness:** 2
**Presentation:** 3
**Significance:** 2
**Originality:** 2
**Overall Recommendation:** 3
**Confidence:** 3

**Summary:**

This paper propose CPD, using the fixed system prompt to estimate a robust
baseline via the median and median absolute deviation, it standardizes user-token entropies and monitors them with a one-sided CUSUM statistic. The resulting detector is model-agnostic, training-free, operates online, and localizes the onset of adversarial suffixes. On a benchmark of 724 optimization-based suffix attacks (GCG, AutoDAN, AdvPrompter) and 765 benign prompts from a TyDiQA+OpenOrca mixture with controlled post-prefix perplexity, CPD consistently outperforms perplexity baselines; on LLaMA-2-7B it reaches AUROC 0.90 and F1 0.82. Here are the comments:

**Compliance With Llm Reviewing Policy:**

Affirmed.

**Final Justification:**

The author partially address my concern, while the need for the external guard is still a weak of this paper. Therefore, I maintain my score.

**Key Questions For Authors:**

# Questions
1. How does model's performace on benign suffixes? For example, when the input has context like "efwefweg@$#$##@...@$#$#$#", will they be detected as adversarial suffixes? Moreover, what about the json or xml format input?

**Limitations:**

Yes

**Strengths And Weaknesses:**

# Strength
1. Taking the adversarial suffix detection as online change-point detection problem is a valid and novel perspective. Beyond that, the CUSUM framework is a well-established method for online change-point detection and fit for this task well.
2. This paper has provided the code of the proposed method, making it easy to reproduce the results.
3. Compared with the currect guardrail systems, CPD can reduce the computational overhead. The method only requires a fixed system prompt and a few hyperparameters.


# weakness
1. Although the new perspective that taking the adversarial suffix detection as online change-point detection problem is valid, the method is still related to the token-level entropy, which is not new. There are many research utilize the entropy/perperlexcity for detection.
2. To achieve the full safety, the paper shows that an external guard is still needed (Section 3.5). Moreover, the paper only focus on the adversarial suffix detection, it's not clear whether the method is effective on other types of (adaptive) jailbreak attacks, such as use the adversarial template to disguise the harmful content.
3. This paper does not include "Qwen" series of models in the experiments, which is a popular and widely used open-source model.

---

> ### Author Rebuttal · Authors · 2026-03-30
>
> We appreciate the questions on scope and robustness, and address each below with new experiments.
>
> On novelty: prior work used global or windowed scalar statistics such as perplexity for adversarial detection (Jain et al., 2023, "Baseline defenses for adversarial attacks"; Alon & Kamfonas, 2023). Our contribution is modelling detection as sequential change-point detection over token streams, both entropy and NLL signals benefit from this formulation. A 2×2 ablation on LLaMA-2-7B confirms the mechanism matters far more than the signal:
>
> | Mechanism | NLL (F1/AUROC) | Entropy (F1/AUROC) |
> |-----------|---------------|-------------------|
> | CUSUM | 0.89 / 0.94 | 0.82 / 0.91 |
> | Window w=1 | 0.72 / 0.75 | 0.69 / 0.70 |
>
> CUSUM improves detection by 13–17 F1 points on either signal. No prior work applies CPD at this granularity. The other aspects are system-prompt calibration via median/MAD (no external training data needed) and joint detection plus token-level localization from one scalar statistic.
>
> On the external guard requirement: we agree that CPD alone is not a complete safety solution, we position it as a lightweight first-stage gate. As shown in Section 3.5, the hybrid CPD+LLaMA Guard pipeline reduces guard invocations by 17–22% while preserving guard-level F1, making full safety coverage practical at reduced compute.
>
> On Qwen models: we added Qwen2.5-7B and Qwen2.5-14B under the same 5-fold CV protocol:
>
> | Model | CPD F1/AUROC | w=1 F1/AUROC |
> |-------|-------------|-------------|
> | Qwen-7B | 0.85 / 0.92 | 0.83 / 0.90 |
> | Qwen-14B | 0.85 / 0.91 | 0.74 / 0.79 |
>
> The method generalizes beyond the LLaMA/Vicuna family.
>
> On benign edge cases (Q1): on pure structured content (JSON, XML, gibberish, code as you described), CPD has very low FPR (1–3%) because these inputs have uniformly high entropy throughout, no sustained shift for CUSUM to accumulate. Windowed PP flags 36–46% of the same prompts because their absolute NLL exceeds the threshold regardless of structure. For prompts where natural language transitions into structured content, CPD's shift sensitivity can produce false positives, a known limitation discussed in Section 5. In deployment, we recommend CPD as a first-stage gate with a downstream classifier (our hybrid gating analysis, Section 3.5), which reduces LLaMA Guard invocations by 17–22% while preserving guard-level F1.

---

> > ### Author Rebuttal · Reviewer_iGSq · 2026-04-03
> >
> > The author partially address my concern, while the need for the external guard is still a weak of this paper. Therefore, I maintain my score.

---

### Official Review · Reviewer_Twhr · 2026-03-12

**Soundness:** 3
**Presentation:** 3
**Significance:** 3
**Originality:** 3
**Overall Recommendation:** 4
**Confidence:** 3

**Summary:**

The paper studies the detection problem for prompt injection -- particularly when the malicious request is appended with a suffix that is optimized for jailbreaking while keeping the overall perplexity low with methods such as AutoDAN. They propose to consider the point-wise token perplexity as a time-series and detect the anomaly.

**Compliance With Llm Reviewing Policy:**

Affirmed.

**Key Questions For Authors:**

Please see above.

**Limitations:**

Please see above.

**Strengths And Weaknesses:**

**Strengths**:

1- The paper reads well and the sections are explained well. The idea of the paper is well-motivated and looks at an important problem where ordinary perplexity filters fail.

2- Their method using CUSUM and how they calibrate the tokens perplexity using the system prompt is sound.

3- The results are promising where the catch the highest number adversarial prompts compared to other methods, while keeping "in-benign" ratio lower -- resulting in better F1 score overall.

4-In Appendix C, they conduct a thorough case study for the benign samples with higher perplexity, where they compare the results for several values of perplexity gap (PP-gap). This is important since the user message do not necessarily have a low perplexity compared to the system prompt (which is used as the calibration baseline)

**Weaknesses**:

1- There is still a hidden assumption (as shown in figure 1) that the adversarial suffix has a slightly higher perplexity that we can exploit and detect. Whereas the main claim of the paper is that they can detect "fluent" adversarial prompts.

2- That said, I think a more detailed comparison with other method is needed. I cannot understand why a windowed perplexity filter with a threshold is outperformed by the suggested method where both of the focus on accumulated point-wise anomaly, e.g., there is a part of the adversarial prompt that has higher perplexity.

3- As I understand the method uses the all of the attacks (GCG, advprompter,  Autodan) is the cross-validation and fits to their accumulated anomaly. This does not address any other attacks that are not seen and can have lower perplexity and different patterns, or having the adversarial part anywhere but the end of the prompt.

4- I think an experiment when one of the attacks is left for choosing the threshold can be helpful with my previous argument.

5- Lastly, what happens if the adversarial part is at the beginning of the sequence where there is not shift between the user bast task and the adversarial part?

---

> ### Author Rebuttal · Authors · 2026-03-30
>
> We thank the reviewer for the detailed suggestions. We ran new experiments to address each point.
>
> 1) On the hidden PP assumption: CPD does not require the suffix to have higher global perplexity. It detects a sustained *shift* in the token-level entropy stream, not an absolute level difference. Our PP-gap ablation (Table 5 in Appendix C, where we vary the benign-adversarial perplexity overlap) demonstrates this: at α=3, benign prompts have *higher* global PP than adversarial ones, yet CPD improves (AUROC 0.84→0.90) while the best windowed PP degrades (0.81→0.72).
>
> 2) On why CPD outperforms windowed PP: smaller windows consistently perform better for WPP — w=5 outperforms w=10/15/20 and w=1 is the strongest overall — which is consistent with the boundary-smearing effect. Larger windows dilute the adversarial signal when they straddle the suffix onset (Table 2, our locality breakdown: 20–40% of WPP triggers straddle the boundary vs 0% for CPD). But even w=5 and w=1, which have minimal or no boundary issue, still fall short of CPD. The limitation is that windowed methods score each window independently with no memory across positions. CPD accumulates evidence over consecutive tokens and resets when the evidence fades, which lets it detect sustained shifts that independent window scoring misses. The 2×2 ablation bears this out: CUSUM outperforms the best windowed baseline by 13–17 F1 points on the same signal.
>
> 3) On unseen attack generalization: we ran leave-one-attack-out (LOAO) experiments where we train the detection threshold on N−1 attack families and evaluate on the held-out family plus benign prompts. We compare against w=1 as it is the strongest windowed baseline overall. We report AUROC as it is threshold-free and thus better suited for cross-attack evaluation where the optimal operating point may differ per attack family:
>
> | Protocol | Held-out pool | CPD AUROC | w=1 AUROC |
> |----------|--------------|-----------|-----------|
> | LOAO-3 | {GCG, AutoDAN, AdvPrompter} | 0.91±0.08 | 0.77±0.13 |
> | LOAO-4 | + BEAST (beam-search fluency attack) | 0.89±0.08 | 0.74±0.13 |
> | LOAO-5 | + AutoDAN-HGA (stealthy genetic variant) | 0.91±0.08 | 0.79±0.15 |
>
> In LOAO-3, the hardest held-out case is AdvPrompter (CPD AUROC 0.89 vs w=1 0.63), the most fluency-optimized family, yet CPD still generalizes well.
>
> 4) We also extended Table 1 by adding BEAST (Sadasivan et al., 2024), a beam-search fluency attack, giving 4 attack families across all 6 models:
>
> | Model | CPD F1/AUROC | Best WPP F1/AUROC |
> |-------|-------------|------------------|
> | LLaMA-2-7B | 0.82 / 0.90 | 0.74 / 0.74 (w=1) |
> | LLaMA-2-13B | 0.82 / 0.90 | 0.71 / 0.73 (w=5) |
> | Vicuna-7B | 0.76 / 0.83 | 0.76 / 0.83 (w=1) |
> | Vicuna-13B | 0.81 / 0.86 | 0.76 / 0.83 (w=1) |
> | Qwen-7B | 0.87 / 0.93 | 0.81 / 0.88 (w=1) |
> | Qwen-14B | 0.87 / 0.91 | 0.74 / 0.75 (w=1) |
>
> CPD outperforms the best windowed baseline on F1 for 5 of 6 models at `k=0`.
>
> 5) On prefix-position attacks: this is a valid limitation that we acknowledge. We limit the scope of this submission to optimization-based suffix attacks, which are the dominant class of automated attacks. Prefix and indirect prompt injection have different sequential structure and we will state this explicitly as future work.

---

> > ### Author Rebuttal · Reviewer_Twhr · 2026-04-04
> >
> > I thank the authors for their additional experiments. I still have few concerns:
> >
> > 1- The method is designed for PP anomaly. If the prompt is really fluent (e.g., generated by an LLM-based method such as PAIR) the method cannot detect it. This is not necessarily a shortcoming, as long as the authors tone down their claims and state that their method is an effective perplexity filter.
> >
> > 2- Their advantage over window-based methods lies in the assumption that the adversarial part is at the end of the prompt. For instance, if this part was in the middle, their method (tailored to detecting adversarial suffixes) would be over-performed by window-based methods.
> >
> > 3- As stated, they only focus on adversarial suffixes, and not any other possible spots for the adversarial parts.

---

> > > ### Author Response · Authors · 2026-04-07
> > >
> > > Thank you for the follow up. We want to clarify the scope and claims.
> > > 1. Our method is not a global perplexity threshold. It uses token-level NLL/entropy over the sequence and detects sustained shifts with CUSUM. So it is more than a standard PP filter.
> > > 2. and 3. We agree the current experiments are focused on suffix attacks. We will state this more clearly and avoid broad claims beyond that setting.

---

### Official Review · Reviewer_Xsxz · 2026-03-12

**Soundness:** 3
**Presentation:** 3
**Significance:** 4
**Originality:** 2
**Overall Recommendation:** 5
**Confidence:** 3

**Summary:**

Introduces a method to detect / localize adversarial suffixes in deployed LLMs by tracking the entropy over stream of token logits. This approach demonstrates benefits over perplexity and window perplexity defenses. It also reduces the cost of using more expensive defenses like LlamaGuard by only calling them when needed.

**Compliance With Llm Reviewing Policy:**

Affirmed.

**Key Questions For Authors:**

1. Was there any attempt to combine both perplexity and entropy metrics into the overall algorithm? Intuitively, both perplexity and entropy reflect different aspects of KL divergence (KL(q || p) = H(q, p) - H(q)).
2. The questions in 198 / 209 could be better formatted to make it less confusing
3. Since Table 3 includes precision and recall, it would be better for table 4 to also include these same metrics to make comparisons easier.

**Limitations:**

yes

**Strengths And Weaknesses:**

Strengths:
- The intuition is fairly straight forward — adversarial attacks induce sustained changes in LLM entropy due to being asked to perform tasks / provide information outside of alignment distribution.
- The method is cheap and pragmatic — it is easy to imagine how this would be added as a very small layer right before the sampling and after the LM head.
- It allows for usage of stronger, more specialized defenses like Llama guard to be called after to verify while saving compute by calling them less frequently.

Weaknesses
- There is not as much theoretical justification behind why this works. I think there should be some deeper reason as to why it works, at least in this case — maybe in terms of the adversarial suffix taking the LLM out of distribution of what it is trained on?

---

> ### Author Rebuttal · Authors · 2026-03-30
>
> Thank you for the encouraging feedback. We address your questions below.
>
> On the theoretical justification: as we discuss in Section 2.1 (Eq. 1), higher next-token entropy gives an attacker more plausible token choices to optimize over without sacrificing fluency. This means fluency-optimized attacks like AutoDAN (Zhu et al., 2023) inherently produce tokens where the model's uncertainty is elevated. The shift is not isolated spikes but a sustained change after the suffix onset. CUSUM (Section 2.3) is the classical tool for detecting persistent mean shifts in sequential data, which is why we adopt it here. A 2×2 ablation (CUSUM vs window × NLL vs entropy) confirms that the detection gain comes from this sequential accumulation mechanism (13–17 F1 points) rather than the choice of signal (3–7 F1 points).
>
> On combining entropy and perplexity: we tested variants that run CUSUM on both NLL and entropy simultaneously and combine the two statistics (via max or sum) on LLaMA-2-7B:
>
> | Method | F1 | AUROC |
> |--------|-----|-------|
> | CUSUM-NLL | 0.89 | 0.94 |
> | CUSUM-Entropy | 0.82 | 0.91 |
> | Combined max | 0.88 | 0.94 |
> | Combined sum | 0.86 | 0.93 |
>
> The combined variants did not exceed the best single-signal CPD.
>
> On your presentation suggestions: we revised Table 4 to include precision and recall alongside F1. At the F1-optimal operating point:
>
> | Guard | Detector | Precision | Recall | F1 | Calls Saved |
> |-------|----------|-----------|--------|-----|-------------|
> | LG1 | CPD Online | 0.80 | 0.85 | 0.82 | 22.3% |
> | LG1 | WPP5 | 0.78 | 0.85 | 0.81 | 8.9% |
> | LG2 | CPD Online | 0.81 | 0.67 | 0.73 | 16.7% |
> | LG2 | WPP15 | 0.81 | 0.67 | 0.73 | 2.3% |
>
> CPD gating preserves guard precision while saving 17–22% of calls. We will also reformat the three research questions (lines 198/209) as a numbered list for clarity.

---

> > ### Author Rebuttal · Reviewer_Xsxz · 2026-04-05
> >
> > I thank the authors for the additional results. I maintain my positive view of the submission.

---

### Decision · Program_Chairs · 2026-04-30

**Decision:**

Accept (regular)

**Comment:**

This paper introduces a pragmatic, training-free method to detect adversarial prompt suffixes by framing token-level perplexity and entropy monitoring as an online change-point detection problem using CUSUM.

The reviewers appreciated the method's simplicity and its practical utility as a lightweight, first-stage safety filter capable of saving significant compute. During the rebuttal, the authors effectively resolved the primary reviewer concerns:

* Source of Gains: A comprehensive 2x2 ablation confirmed that the performance improvements stem from the sequential accumulation mechanism (CUSUM) rather than the specific choice of signal (NLL vs. entropy).

* Generalization: The authors successfully extended the evaluation to newer architectures (Qwen-7B/14B) and demonstrated strong generalization to unseen attacks (e.g., BEAST) via leave-one-attack-out cross-validation.

* Experimental Rigor: By rerunning the experiments without the controversial perplexity multiplier ($\alpha=1$), the authors proved the method still outperforms windowed baselines in standard, unmodified settings.

While the scope of the defense is limited to suffix-based attacks (which the authors should explicitly note as a limitation in the camera-ready version), the approach is empirically sound, highly practical, and provides a valuable contribution to efficient LLM safety pipelines.